# *LitExplorer*: A Plugin for Enhancing Training-Free Diffusion Alignment in Efficiency and Diversity

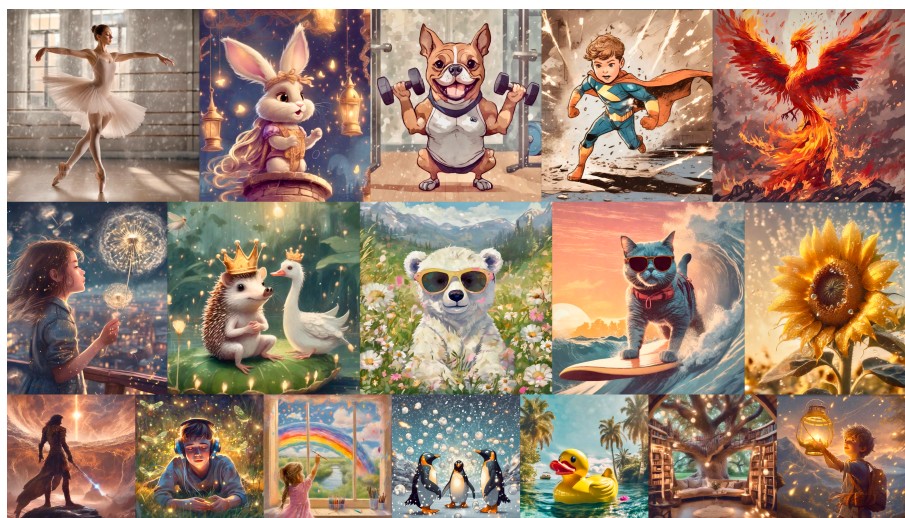

Figure 1: **Showcase of generated samples.** The images from our model achieve a state-of-the-art performance based on SD-XL.

## ABSTRACT

Diffusion models have general generative abilities but struggle to align with specific objectives. Fine-tuning can improve alignment, yet its training cost is often prohibitive. This led to training-free methods that apply objective-guided terms in sampling to bias the distribution toward designated regions, e.g., high-reward areas. However, these methods face two key issues: (1) the strong directional bias narrows the pretrained distribution, and (2) indiscriminate guidance fails to prune redundant signals, hurting both quality and efficiency. To solve the above challenges, we propose *LitExplorer*, a plugin that mitigates distribution collapse and reduces compute. Firstly, we adopts an Inheritance-Restart exploration mechanism, using probabilistic perturbation to avoid early convergence, while exploration also raises the chance of high-reward trajectories. Then, it balances diversity and fidelity, adding diversity without distribution shift. Second, our Quality–Efficiency arbitration mechanism improves guidance by removing incorrect signals and cuts computation through dynamic early stopping driven by generation completeness and marginal reward gain. Experimental results indicate that the proposed *LitExplorer consistently* achieves superior performance across **12** metrics, encompassing preference, fidelity, diversity, and richness.

## 1 INTRODUCTION

Diffusion models (Ho et al., 2020; Ramesh et al., 2022; Rombach et al., 2022), as one of the most powerful generative frameworks, demonstrate exceptional generative capabilities by training on various large-scale datasets. This ability is reflected not only in high-quality image generation but also

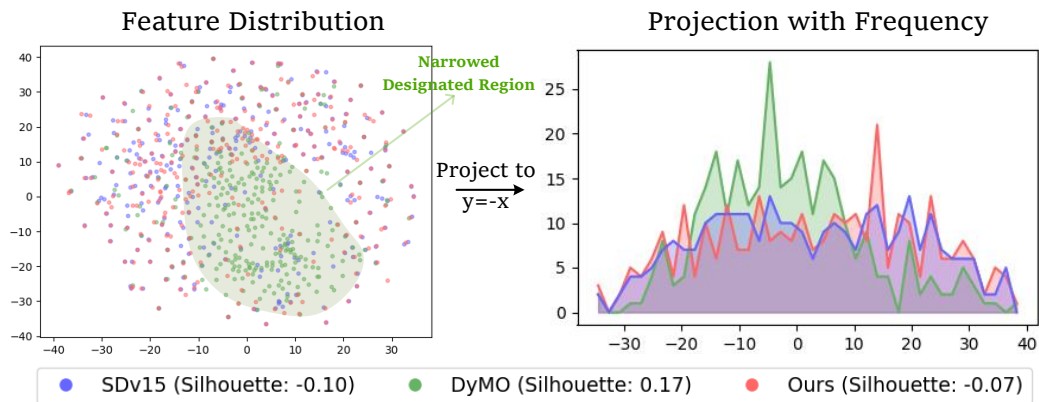

Figure 2: Feature visualization via t-SNE. The features are extracted by a general self-supervised decoder (*i.e.*, DINOv2) from images generated by different methods. The results indicate that existing methods (e.g., DyMO) designate more compact clustering, narrowing the range of the generated distribution, which in turn reduces the diversity of the generated data. This observation is substantiated by the Silhouette Coefficient (Rousseeuw, 1987), a clustering evaluation metric, which demonstrates its convergence and limited diversity. In contrast, our method maintains a broader generative distribution while achieving comparable target reward scores (see Tab. 1). *See Appendix. 12 for results of other guidance methods in this setting.*

in their capacity to cover a broad semantic and stylistic space. However, in practical applications, there is a need not only to generate realistic images but also to ensure that the results align with specific target requirements, such as aesthetic preferences. This need for goal alignment is becoming a core research problem, posing the challenge of how to effectively translate the general capabilities of diffusion models into customized generation (Yeh et al., 2024; Kim et al., 2025).

Early research primarily relied on training strategies (Black et al., 2023; Fan et al., 2024; Yang et al., 2024; Wallace et al., 2024). By fine-tuning diffusion models on specific preference objectives, alignment performance could be significantly improved. However, such methods generally face challenges of high training costs and poor transferability (Kim et al., 2025; Xie & Gong, 2025). Whenever user requirements or objectives change, diffusion models must be fintuned again, which limits their scalability in practical applications. To address this, researchers have begun searching training-free methods (Chung et al., 2022; Kim et al., 2025; Deckers et al., 2024; Tang et al., 2024; Xie & Gong, 2025), which do not modify the diffusion model's weights but instead incorporate external signals during generation to achieve goal-guided results. The advantages of such approaches are evident as they require no additional training costs and can quickly adapt to different objective demands. Consequently, training-free guidance has gradually become one of the crucial directions in diffusion model alignment. More detailed **Related Works** can be found in Appendix A.

However, existing training-free methods still suffer from two typical limitations: (1) current methods suffer from overly narrow generative distributions. To enhance the alignment between generated results and the target objectives, training-free methods need to apply target-guided terms during the sampling process. These methods gradually shift the generative distribution toward specific regions, such as high-reward subspaces. Although these approaches effectively enhance alignment, their reliance on a consistent guidance direction easily leads to reward hacking (Tang et al., 2024). In turn, causing a rapid narrowing of the pre-trained distribution, ultimately resulting in reduced image diversity (Ho & Salimans, 2022; Tang et al., 2024); (2) The issue of inefficient guidance cannot be overlooked. Current methods apply guidance terms indiscriminately throughout the entire generation process, lacking the ability to identify and manage ineffective or redundant signals. This not only results in unnecessary computational waste but may also introduce noise interference due to ineffective guidance, ultimately compromising both alignment quality and generation efficiency.

To address the above challenges, we propose *LitExplorer*, an enhanced plugin compatible with existing training-free alignment methods. Specifically, *LitExplorer* first introduces an Inheritance-Restart exploration mechanism that prevents premature convergence of generation trajectories to a single mode through probabilistic perturbation and path screening. Simultaneously, it adaptively balances

Figure 3: Overall framework of the proposed plug-and-play *LitExplorer*.

exploration and fidelity based on the generation progress, which mitigates diversity loss caused by excessive guidance without deviating from the original distribution (we provide visual evidence in Fig. 2). By incorporating this adaptive trade-off, the sampling process encourages exploratory potential in early stages to enhance diversity. It gradually converges in later stages to maintain the pre-trained distribution, thereby balancing creativity and reliability. Moreover, the progressive incorporation of exploration during the generation process increases the likelihood of discovering high-reward trajectories. Secondly, to improve guidance efficiency, *LitExplorer* employs a Quality-Efficiency screening mechanism to filter out ineffective signals, ensuring that each guidance step contributes meaningfully. Additionally, by integrating an adaptive factor and considering the diminishing marginal utility of rewards, our method dynamically cuts guidance in later generation stages to avoid redundant optimization. Quality-Efficiency mechanism effectively reduces computational overhead while maintaining alignment quality. Experiments demonstrate that while ensuring leading performance in both fidelity and alignment objectives, *LitExplorer* effectively improves generative diversity and outperforms existing baseline methods in computational efficiency. Our contributions are summarized as follows:

- We propose Inheritance-Restart exploration mechanism to enhance diversity.
- We introduce a Quality-Efficiency arbitration mechanism to drop invalid signals, ensuring better guidance and lower cost.
- We achieve two trade-offs: Diversity and Fidelity by generation progress. Further, Quality and Efficiency by marginal reward gain.

## 2 METHOD

*LitExplorer* is a plugin that enhances training-free methods from exploration and efficiency perspectives. First, in Sec. 2.1, we provide the necessary preliminaries for *LitExplorer*. In Sec. 2.2, we elaborate on the Inheritance-Restart exploration mechanism of *LitExplorer*, achieving an adaptive trade-off between Diversity and Fidelity. Subsequently, in Sec. 2.3, we introduce a Quality–Efficiency arbitration mechanism, which ensures guidance for quality in the early generation stages and efficiency in the later stages. *LitExplorer* framework is shown in Fig. 3.

### 2.1 REWARD-GUIDED DIFFUSION ALIGNMENT

#### 2.1.1 GENERATIVE PROCESS OF DIFFUSION MODELS

The discrete-time generative process (or reverse process) aims to produce a data sample by iteratively denoising a latent variable. This process requires two key components: a predefined noise schedule $\{\beta_t\}_{t=0}^T$ and a well-trained score function $\mathbf{s}_\theta(\mathbf{x}_t, t)$. For conciseness, we define $\alpha_t = 1 - \beta_t$ and $\bar{\alpha}_t = \prod_{i=1}^t \alpha_i$. The generation starts by sampling an initial latent variable from a standard normal distribution: $\mathbf{x}_T \sim \mathcal{N}(\mathbf{0}, \mathbf{I})$ Then, for each timestep $t$ from $T$ down to $0$, the sample is iteratively refined using the following update rule:

$$\mathbf{x}_{t-1} = \frac{1}{\sqrt{\alpha_t}} \left( \mathbf{x}_t + \beta_t \mathbf{s}_\theta(\mathbf{x}_t, t) \right) + \sqrt{\beta_t} \mathbf{z}_t \tag{1}$$

where $\mathbf{z}_t \sim \mathcal{N}(\mathbf{0}, \mathbf{I})$. The final output after $T$ steps is the generated sample $\mathbf{x}_0$. For convenience, we also define a function $f_\theta(\mathbf{x}_t, t)$ that provides a direct estimate of the clean sample $\hat{\mathbf{x}}_0$ from any noisy intermediate $\mathbf{x}_t$, $\mathbf{x}_{0|t} = f_\theta(\mathbf{x}_t, t) = \frac{1}{\sqrt{\bar{\alpha}_t}}(\mathbf{x}_t + (1 - \bar{\alpha}_t)\mathbf{s}_\theta(\mathbf{x}_t, t))$(Chung et al., 2022; Bansal et al., 2023; Xie & Gong, 2025).

### 2.1.2 ITERATIVE REFINEMENT WITH GRADIENT GUIDANCE

To guide the generation process towards a specific objective, we can incorporate the gradient of a reward function $\mathcal{R}$. The reward function is defined in the data domain (i.e., it operates on the estimated clean sample $\mathbf{x}_{0|t}$).

The guidance is injected at each step by first perturbing the current sample $\mathbf{x}_t$ in the direction of the reward gradient. Specifically, before applying the update in Eq. 1, we compute a guided sample $\mathbf{x}_t$:

$$\mathbf{x}_t \leftarrow \mathbf{x}_t + w\nabla_{\mathbf{x}_t}\mathcal{R}(f_\theta(\mathbf{x}_t, t)), \tag{2}$$

where $w$ is a guidance scale factor, $\mathcal{R}(\cdot)$ denotes the reward function for inference-time guidance. The above process could be seen as one step of optimization. For simplicity, we can define optimization with $m$ steps of Eq. 2 as

$$\tilde{\mathbf{x}}_t = g(\mathbf{x}_t, \mathcal{R}, m). \tag{3}$$

This guided sample $\tilde{\mathbf{x}}_t$ then replaces $\mathbf{x}_t$ in the denoising update rule:

$$\mathbf{x}_{t-1} = \frac{1}{\sqrt{\alpha_t}}(\tilde{\mathbf{x}}_t + \beta_t \mathbf{s}_\theta(\tilde{\mathbf{x}}_t, t)) + \sqrt{\beta_t}\mathbf{z}_t. \tag{4}$$

## 2.2 INHERITANCE–RESTART EXPLORATION MECHANISM

To mitigate the issue of narrowed generative distribution caused by overly concentrated guidance direction in training-free methods, *LitExplorer* equips an Inheritance and Restart exploration mechanism. This mechanism selectively introduces exploratory supplementary terms based on the generation progress, thereby preventing premature convergence of generative trajectories to a single mode and increasing the probability of discovering high-reward paths.

### 2.2.1 EXPLORATION SUPPLEMENT VARIABLE

During the generation step $t$, we introduce exploration supplement variable $\boldsymbol{\epsilon}_t^{(i)}$, which allows the generated trajectories to maintain a richer distribution. Specifically, we adopt Monte Carlo sampling to obtain $n$ candidates $\boldsymbol{\epsilon}_t^{(i)} \sim \mathcal{N}(0, \sigma_t^2 I)$, $i = 1, \ldots, n$. After incorporating $\boldsymbol{\epsilon}_t^{(i)}$, the original latent variable $\mathbf{x}_t$ without exploration is updated into a set $\{\hat{\mathbf{x}}_t^{(i)}\}$:

$$\hat{\mathbf{x}}_t^{(i)} = \mathbf{x}_t + \boldsymbol{\epsilon}_t^{(i)}. \tag{5}$$

Subsequently, each candidate $\hat{\mathbf{x}}_t^{(i)}$ is mapped to its predicted reconstruction $\hat{\mathbf{x}}_{0|t}^{(i)} = f_\theta(\hat{\mathbf{x}}_t^{(i)}, t)$, and is evaluated via the objective reward function $\mathcal{R}(\hat{\mathbf{x}}_{0|t}^{(i)})$. After $m$ rounds of iteration based on Eq. 3, we obtain a final set of candidates $\{\tilde{\mathbf{x}}_t^{(i)}\} = g(\{\hat{\mathbf{x}}_t^{(i)}\}, \mathcal{R}, m)$. We then define the optimal generation trajectory policy as:

$$i^\star = \arg\max_{i \in \{1,\ldots,n\}} \mathcal{R}(\tilde{\mathbf{x}}_{0|t}^{(i)}), \quad \mathbf{x}_t^\star = \tilde{\mathbf{x}}_t^{(i^\star)}. \tag{6}$$

This process picks the optimal candidate $\mathbf{x}_t^\star$ as the state of generation step $t$, which is adopted for the following generations. In summary, the injection of the exploration supplementary term expands the local search space. Next, the selection policy is employed to pick the optimal generation trajectory with the highest alignment potential, thereby achieving increased diversity while maximizing alignment rewards.

### 2.2.2 INHERITANCE–RESTART TECHNIQUE

In Sec. 2.2.1, the exploration supplementary term expands the generation space. Building upon this, we apply an Inheritance-Restart technique to this term. This technique dynamically regulates

exploration behavior based on the target reward improvement, preserving high-quality explorations while eliminating ineffective perturbations. Consequently, the exploration term is responsible for diversity enhancing, while the Inheritance-Restart technique ensures that the exploration consistently improves the target reward. Specifically, the reward changing for the supplementary term at generation step $t$ is defined as:

$$\Delta\mathcal{R}_t = \mathcal{R}(\mathbf{x}_{0|t}^\star) - \mathcal{R}(\mathbf{x}_{0|t}), \tag{7}$$

**(1)** If $\Delta\mathcal{R}_t > 0$, it indicates that the exploration still yields a positive gain for the alignment reward. In this case, the latent change from the previous step is carried over to the next step $t-1$, where $\boldsymbol{\epsilon}_{t-1} = \mathbf{x}_t^\star - \mathbf{x}_t$. Under this inheritance strategy, the exploration candidate set contains only this inherited signal, i.e., setting $n = 1$ in Eq. 5, and optimization continues via Eq. 3.

**(2)** If $\Delta\mathcal{R}_t < 0$, it indicates that the exploration gain becomes saturated or starts to decline. In this case, the restart mechanism is activated: $n$ exploration candidates are resampled in the next step $t-1$ as Eq. 5, and the exploration term is updated through the optimization in Eq. 3 and the selection step in Eq. 6.

The inheritance-restart technique uses the reward as a criterion to enhance exploration diversity and efficiency while ensuring benefits to the target alignment.

### 2.2.3 DIVERSITY AND FIDELITY ADAPTIVE TRADE-OFF

Diffusion models face the dual demands of diversity and fidelity. Diversity expands the coverage of potential outputs, while fidelity ensures the results are credible and semantically consistent with the conditioning. However, these two objectives often exhibit a trade-off during the generation process, making their careful balance crucial for achieving high-quality results (Dhariwal & Nichol, 2021). To address this, we propose an adaptive coordination technique that dynamically adjusts the emphasis based on the generation progress: it prioritizes exploration in the early stages to foster diversity and gradually increases the strength of regularization in the later stages to ensure fidelity. Unlike approaches that use predefined static parameters, the adaptation technique can flexibly accommodate the dynamic nature of the generation process.

To this end, we construct a Control Network*(This network outputs a simple noise scorer, analogous to scoring models such as PickScore and Aesthetic Score, and is therefore used as a pretrained model.Experimental support is provided in Tab. 10, with details in Appendix C.3.)*, which is designed to assess the denoising progress of the intermediate state $x_t$ in real-time during the generation process. Specifically, the control network is defined as a mapping function: $h_{\theta_p} : \mathbf{x}_t \mapsto p_t$, where $\mathbf{x}_t \in \mathbb{R}^d$ denotes the state at generation step $t$, $p_t \in [0, 1]$ represents the probability that the current state is close to the clean image $\mathbf{x}_0$. This pre-trained network is obtained by minimizing a binary cross-entropy loss as the supervision signal (Appendix B.1 for more details). The control network can provide an adaptive factor $p_t$ for any intermediate state $\mathbf{x}_t$ during inference, where $p_t = h_{\theta_p}(\mathbf{x}_t)$. This factor reflects the degree of generation progress of the current state. We utilize $p_t$ as an adaptive weighting to balance exploration diversity and generation fidelity. In the first round of optimization, the combined effect of the exploration and alignment signal gradients is written as:

$$\hat{\mathbf{x}}_t = \mathbf{x}_t + \underbrace{(1 - p_t) \cdot \boldsymbol{\epsilon}^{(i)} - p_t \cdot \nabla_{\mathbf{x}_t} L_2(\hat{\mathbf{x}}_t, \mathbf{x}_t)}_{\text{Diversity-Fidelity Trade-off}} + \nabla_{\mathbf{x}_t} \mathcal{R}(\mathbf{x}_{0|t}), \tag{8}$$

where $\boldsymbol{\epsilon}^{(i)}$ is the exploration supplement variable and $\nabla_{\mathbf{x}_t} \mathcal{R}(\mathbf{x}_{0|t})$ represents the reward-based gradient guidance. We introduce L2 regularization to constrain the exploration scope. Then, based on the iteration defined in Eq. 3, we iteratively introduce guidance to obtain the final state $\tilde{\mathbf{x}}_t = g(\hat{\mathbf{x}}_t, \mathcal{R} - L_2, m)$.

In summary, the adaptive trade-off technique enables the encouragement of exploratory diversity in the early stages of generation. In the later stages, the continuous increase of the regularization constraint prompts the generation process to transition from exploration to stable convergence gradually, thereby ensuring high fidelity of the results. Through this design, *LitExplorer* achieves an adaptive trade-off between creativity and stability.

## 2.3 QUALITY–EFFICIENCY ARBITRATION MECHANISM

### 2.3.1 GUIDANCE SCREENING

Existing methods apply guidance signals indiscriminately throughout the entire sampling process, which can lead to ineffective guidance and undermine alignment effectiveness. To address this, *LitExplorer* introduces a guidance signal selection technique that dynamically discriminates the signal's utility. Let the optimal guidance signal generated at step $t$ be $\gamma_t^{(i^\star)}$, which represents the reward gradient corresponding to $\mathbf{x}_t^\star$, *i.e.*, the optimal candidate. We then define a selection operator $\mathcal{F}$:

$$\mathcal{F}(\gamma_t^{(i^\star)}) = \{\Delta \mathcal{R}_t > 0\}.$$

With the proposed screening process, a guidance signal is retained only if it contributes to a positive reward gain; otherwise, it is discarded, and the original latent $x_t$ from the base diffusion model is used for the subsequent step. This reward-margin-based selection strategy ensures the quality of the guidance signals for generation.

### 2.3.2 QUALITY–COMPUTE TRADE-OFF

Unlike existing methods that rigidly couple guidance with the generation steps, *LitExplorer* regulates the use of guidance signals through the dual control of target reward marginal gain and generation progress $p_t$. This approach aims to reduce computational costs while ensuring alignment quality. Guidance is stopped when it is insignificant to reward improvement and the denoising meets the required threshold, thereby avoiding redundant costs and preventing over-guidance.

We first denote the step $t$ reward $\mathcal{R}_t(\cdot)$ as $r_t$. Then, the smoothed trend of reward gain is defined as:

$$\widetilde{\Delta r_t} = (1 - \phi)\, \widetilde{\Delta r_{t+1}} + \phi\, (r_t - r_{t+1}), \tag{9}$$

where $\phi \in (0, 1]$. If and only if $p_t = 1$ and $\widetilde{\Delta r_t} \leq \delta_r$, it can be interpreted that the reward return margin has been reached and the early stopping is triggered, where $\delta_r \geq 0$ is the minimal tolerance gain. We define the early-stopping indicator function as

$$\mathrm{E}_{stop}(t) = \mathbb{I}\Big[p_t = 1\, \wedge\, \widetilde{\Delta r_t} \leq \delta_r\Big] \in \{0, 1\}.$$

When $\mathrm{E}_{stop}(t) = 1$, no exploration or guidance term is added. If $\mathrm{E}_{stop}(t) = 0$, the exploration and guidance proceed as usual. This criterion ensures early stopping is triggered only when both conditions are met: $p_t$ guarantees the state is "sufficiently clean", while $\widetilde{\Delta r_t}$ ensures that 'further exploration guidance yields no significant reward improvement'. By requiring simultaneous satisfaction of both conditions, this technique eliminates redundant guidance in the later stages of generation without compromising target alignment quality, thereby reducing computational cost.

## 3 EXPERIMENTS

### 3.1 EXPERIMENTAL SETUP

**Datasets.** In this paper, we deploy Pick-a-pic (Kirstain et al., 2023) and HPSv2 (Wu et al., 2023) datasets as our basic test bed. Specifically, we randomly pick 500 prompts from Pick-a-pic validation set and 500 prompts from HPSv2 photorealistic set, since they represent distinct stylistic categories. Namely, the prompts in Pick-a-Pic are relatively complex, abstract, and surreal, whereas those in HPSv2 are more realistic and closely aligned with the real world.

**Optimization Objectives and Metrics.** In this paper, we deploy three reward functions as the optimization objectives, that is, PickScore (Kirstain et al., 2023) (PS), Aesthetic Score (Schuhmann et al., 2022) (AES), and ImageReward (Xu et al., 2023a) (IR). To quantitatively evaluate the generative performance as extensively as possible, we introduce 4 evaluating dimensions with 13 distinct metrics, including: **Aesthetic Preference:** AES, PS, IR, and HPSv2 (Wu et al., 2023). **Image Fidelity:** ClipScore (Radford et al.) (Clip), Fréchet Inception Distance (FID) (Heusel et al., 2017), and improved F1 Score (iFS) (Kynkäänniemi et al., 2019). **Generative Diversity:** LPIPS (Zhang et al., 2018), TCE (Ibarrola & Grace, 2024), and Inception Score (Salimans et al., 2016) (IS). **Compositional Richness:** NIQE (Mittal et al., 2012b), BRISQUE (BRI) (Mittal et al., 2012a), and Spectual

Table 1: Main comparisons with SoTA (**12** in total). All metrics are obtained with SDv15 as backbone, Pick-a-pic as prompt set, and PickScore as reward for guidance. *Orange denotes gradient-based training methods. Green denotes reinforcement-learning-based training methods. Blue denotes training-free methods.*

| Method | Preference | | | | Fidelity | | | Diversity | | | Richness | | | #Top2 |
|---|---|---|---|---|---|---|---|---|---|---|---|---|---|---|
| | PS↑ | AES↑ | IR↑ | HPS↑ | FID↓ | CLIP↑ | iFS↑ | LPIPS↑ | IS↑ | TCE↑ | BRI↓ | NIQE↓ | SE↑ | |
| SDv15 | 20.48 | 5.412 | 0.181 | 0.262 | - | 0.243 | - | **0.654** | **23.79** | 38.05 | 18.66 | 5.401 | 11.26 | 3 |
| Diff-DPO | 20.87 | 5.551 | 0.443 | 0.271 | 109.1 | **0.244** | 0.795 | 0.639 | 22.77 | 39.20 | 15.12 | 4.463 | 11.02 | 1 |
| Diff-KTO | 20.83 | 5.585 | 0.599 | 0.272 | 101.3 | 0.240 | 0.801 | 0.634 | 22.70 | 39.12 | 26.25 | 4.361 | 11.22 | 0 |
| SPO | 20.76 | 5.613 | 0.282 | 0.218 | 78.76 | 0.241 | 0.855 | 0.649 | 23.18 | 39.32 | 25.67 | 4.103 | 11.09 | 1 |
| DRaFT | 22.52 | 5.697 | 0.779 | 0.271 | 84.47 | 0.231 | 0.856 | 0.572 | 20.10 | 37.85 | 35.64 | 5.336 | 10.43 | 2 |
| DDPO | 21.79 | 5.704 | 0.196 | 0.212 | 147.5 | 0.242 | 0.539 | 0.629 | 20.01 | 39.18 | 12.46 | 4.328 | 11.74 | 1 |
| DPOK | 20.97 | 5.661 | 0.5828 | 0.272 | 99.30 | 0.242 | 0.820 | 0.641 | 22.52 | 39.35 | 12.28 | 4.608 | 11.23 | 2 |
| DNO | 20.87 | 5.479 | 0.425 | 0.271 | 79.96 | 0.239 | 0.851 | 0.591 | 19.94 | 38.88 | 17.23 | 4.757 | 11.07 | 0 |
| TTScale | 22.14 | 5.636 | 0.619 | 0.272 | **66.93** | 0.240 | 0.863 | 0.652 | 23.73 | 38.90 | 15.05 | 4.447 | 11.28 | 2 |
| FKSteer | 22.56 | 5.696 | 0.539 | 0.278 | 73.83 | 0.243 | 0.868 | 0.650 | 23.76 | 39.42 | 13.93 | 4.092 | 11.59 | 2 |
| DAS | 22.34 | 5.658 | 0.632 | **0.283** | 95.42 | 0.240 | 0.844 | 0.625 | 21.01 | 39.19 | 14.36 | 4.631 | 11.14 | 1 |
| DyMO | 22.79 | 5.694 | 0.709 | 0.279 | 90.56 | 0.239 | 0.849 | 0.627 | 20.97 | 39.29 | 16.18 | 4.545 | 11.29 | 1 |
| Ours | **22.91** | **5.707** | **0.812** | 0.280 | 71.36 | 0.242 | **0.891** | **0.654** | 23.71 | 39.37 | **10.92** | **2.886** | **12.07** | **11** |

Table 2: Results on HPSv2 dataset.

| Method | IR | PS | CLIP | LPIPS | TCE | NIQE↓ |
|---|---|---|---|---|---|---|
| SDv15 | 0.231 | 21.57 | **0.249** | 0.643 | 39.62 | 4.656 |
| Diff-DPO | 0.571 | 22.03 | 0.245 | 0.619 | 37.97 | 4.471 |
| SPO | 0.300 | 21.58 | **0.249** | 0.595 | 39.37 | 4.172 |
| DDPO | 0.678 | 22.26 | 0.245 | 0.607 | 38.96 | 3.973 |
| DNO | 0.446 | 21.79 | 0.246 | 0.575 | 39.14 | 4.523 |
| DAS | 0.684 | 23.09 | 0.246 | 0.653 | 39.31 | 4.133 |
| DyMO | 0.805 | 23.46 | 0.246 | 0.610 | 39.57 | 4.088 |
| Ours | **0.872** | **23.79** | 0.247 | **0.671** | **39.92** | **2.595** |

Table 3: Results with advanced SD models.

| Method | IR | PS | CLIP | LPIPS | TCE | NIQE↓ |
|---|---|---|---|---|---|---|
| SD-XL | 0.715 | 21.45 | 0.236 | 0.675 | 40.18 | 4.553 |
| Diff-DPO | 1.010 | 22.05 | 0.236 | 0.651 | 39.45 | 4.440 |
| SPO | 1.179 | 22.81 | 0.232 | 0.556 | 40.42 | 3.900 |
| SDv35 | 1.193 | 21.90 | **0.238** | 0.621 | 39.42 | 4.873 |
| DNO | 0.924 | 22.58 | **0.238** | 0.581 | 39.34 | 4.783 |
| DAS | 1.171 | 23.07 | 0.237 | 0.594 | 39.36 | 5.365 |
| DyMO | 1.079 | 24.34 | 0.233 | 0.609 | 39.55 | 4.510 |
| Ours | **1.201** | **24.56** | 0.236 | **0.679** | **40.99** | **3.573** |

Entropy (SE) (Liu et al., 2014). Details of each metric can be found in Appendix.

**Baselines.** For comprehensive comparison, we introduce three types of State-of-The-Art (SoTA) diffusion-based methods, including: **Gradient-based Training:** Diff-DPO (Wallace et al., 2024), Diff-KTO (Li et al., 2024), SPO (Liang et al., 2024), and DRaFT (Clark et al., 2023). **RL-based Training:** DDPO (Black et al., 2023), DPOK (Fan et al., 2023). **Training Free:** DAS (Kim et al., 2025), TTScale (Ma et al., 2025), FKSteer (Singhal et al., 2025), DNO (Tang et al., 2024), and DyMO (Xie & Gong, 2025).

All baselines are strictly reproduced based on their official code and settings within our evaluation benchmark. Stable Diffusion v1.5 (SDv15) is taken as our primary backbone. *More advanced diffusion models such as SD XL1.0 (XL) and SD3 (see Appendix Tab. 11, Fig. 13) are also considered for evaluation and comparison*. All models and results are obtained on a single NVIDIA Tesla H100 GPU. More training details can be found in Appendix.

## 3.2 COMPARISONS WITH SoTA

**Experiments on SDv15.** To extensively demonstrate the effectiveness of *LitExplorer* (LitE), we carefully reproduce and evaluate *12 baselines* with PickScore as the objective on SDv15 and pick-a-pic datasets. Since LitE is an efficient plugin, this section represents deploying LitE to DyMO. Then, we introduce 13 metrics to assess from four dimensions. As shown in Tab.1, our method achieves the best overall performance in all dimensions, achieving Top2 performance in 12 metrics. Moreover, since the prompts in the Pick-a-pic dataset are somehow abstract and surreal, we further conduct an experiment on the photorealistic HPSv2 dataset for comprehensive evaluation. In Tab.2,

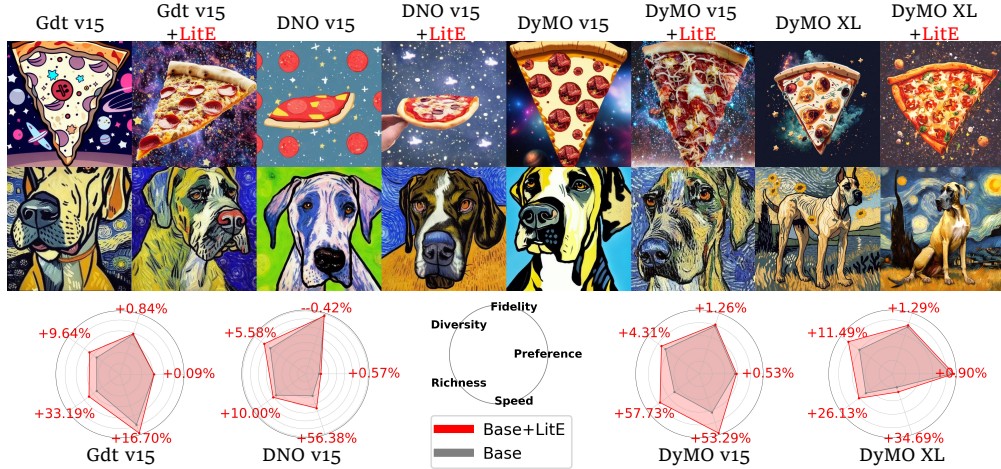

Figure 4: Plug-and-Play effectiveness. While maintaining the Fidelity and Preference, our method significantly enhances Diversity, Richness, and Speed in all cases, such as the "background and pizza filling diversity" in the first row, and "improved Van Gogh style" in the second row.

all methods perform better because realistic objects are more likely to be seen during training and hence easier to generate. Nevertheless, our method still achieves the best results on five metrics and maintains comparable fidelity. The qualitative results could be found in the Appendix. C.9. *In addition, to assess the impact of random seeds on result stability, we compute the coefficient of variation (CoV) of each metric for the training-free baseline across different seeds. These results are presented in Appendix Tab. 9.*

**Comparisons with more advanced SD models.** In Tab. 3, we employ our method and other SoTA on the SD-XL backbone, and introduce the advanced SDv35 as an additional baseline. The results further demonstrate the superiority of our method as an effective plugin.

**Results with alternative optimization targets.** We also conduct experiments by replacing PickScore with Aesthetic score as the optimization target to evaluate the adaptability of *LitExplorer*. Due to the length limitation, the results are shown in Appendix C.9. Briefly, the results indicate our overall superiority on aesthetic preference and generative diversity.

*Computational Cost Comparison. To fairly evaluate the computational cost of each method and verify the efficiency advantages of our approach, we measure and compare the time consumption of all methods under consistent experimental conditions(see Fig 10).*

### 3.3 PLUG-AND-PLAY EFFECTIVENESS

In Fig. 4, we discuss the plugin effectiveness of *LitExplorer*. Specifically, we present side-by-side comparisons of results with and without LitE. Gdt represents backbones with naive Gradient guidance. For simplicity, we choose PS, CLIP, LPIPS, $\frac{1}{\text{NIQE}}$, and image per minute as the representations of Preference, Fidelity, Diversity, Richness, and Speed. The results clearly show that LitE could effectively enhance the diversity of base methods and reduce computational cost while maintaining or even improving other quality dimensions. *To ensure that our integrability remains robust in larger and more advanced models, we further conducted integration experiments on the leading diffusion model SD3, with results shown in Fig. 13 and Tab. 11.*

### 3.4 ABLATION STUDY

**Overall Ablation.** To study the individual effects of the components proposed, we systematically disassembled each component for ablation experiments. *LitExplorer* aims to balance the trade-offs between diversity-fidelity and quality-efficiency, and thus can be divided into four categories based on the objectives. Specifically, Diversity (Dive) includes directly adding exploration and adding Monte Carlo exploration. Fidelity (Fide) refers to L2 constraints for regularization. Efficiency (Effi) includes the I&R and early stopping. Guidance Screening aims to enhance Quality (Qual). Then,

Table 4: Ablation study. For each component, ESV and M-ESV represent exploration supplement variable and with Monte Carlo sampling. I&R and EarS denote Inheritance-Restart technique and Early Stopping. GuiS denotes Guidance Screening.

| Var | Dive | | Fide | Effi | | Qual | IR | PS | CLIP | LPIPS | TCE | NIQE↓ | Time↓ | #Top2 |
| | ESV | M-ESV | L2 | I&R | EarS | GuiS | | | | | | | | |
|---|---|---|---|---|---|---|---|---|---|---|---|---|---|---|
| Base | | | | | | | 0.709 | 22.79 | 0.239 | 0.627 | 39.29 | 4.454 | 44.92 | 0 |
| V1 | ✓ | | | | | | 0.739 | 22.12 | 0.236 | **0.659** | 39.30 | 2.897 | 46.31 | 1 |
| V2 | ✓ | | ✓ | | | | 0.734 | 22.46 | 0.240 | 0.636 | 39.30 | 3.379 | 51.24 | 0 |
| V3 | | ✓ | ✓ | | | | 0.757 | 22.75 | 0.240 | 0.641 | 39.32 | 2.939 | 67.08 | 0 |
| V4 | | ✓ | ✓ | ✓ | | | 0.746 | 22.73 | 0.241 | 0.640 | 39.29 | 3.039 | 48.56 | 0 |
| V5 | | ✓ | ✓ | ✓ | ✓ | | 0.759 | 22.66 | **0.242** | 0.645 | 39.28 | 3.120 | 40.91 | 2 |
| V6 | | ✓ | ✓ | ✓ | | ✓ | 0.774 | **22.95** | 0.241 | 0.653 | 39.35 | **2.847** | 44.73 | 4 |
| Ours | | ✓ | ✓ | ✓ | ✓ | ✓ | **0.812** | 22.91 | **0.242** | 0.654 | **39.37** | 2.886 | **37.28** | 7 |

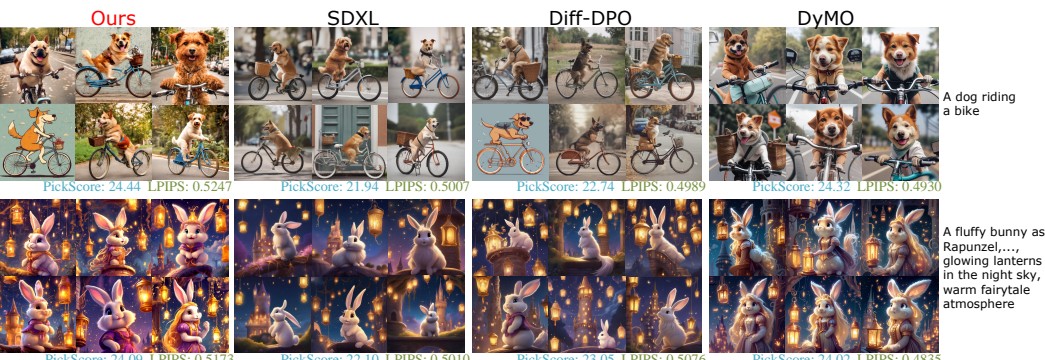

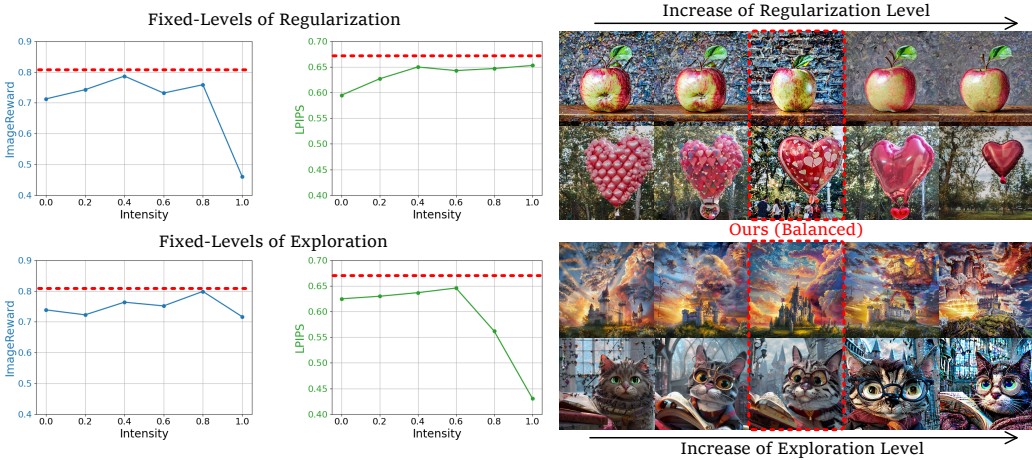

Figure 5: Examples of generative diversity with the same prompt and different seeds.

Figure 6: Comparing our diversity-fidelity adaptive trade-off with different fixed exploration and regularization intensities. The red dotted lines indicate our method.

we conducted detailed ablations in Tab. 4, where the proposed diversity-fidelity tradeoff effectively improves the diversity of generated images while maintaining consistency. Meanwhile, the I&R mechanism and early stopping greatly reduce computational overhead with almost no impact on performance. GuiS improves inference efficiency and performance by removing invalid guidance. *In the Appendix, we further analyze the mutual correlation among different proposed components based on the ablation results.*

**Analysis on Generative Diversity.** The generative diversity is compared among SD-XL, Diff-DPO, DyMo, and Ours by generating images with 40 seeds. As shown in Fig. 5, our method exhibits

improved diversity while maintaining the impressive preference score and precise prompt-image alignment. We have also investigated exploration diversity during inference in Appendix C.10.

**Analysis on Diversity-Fidelity Trade-off.** In Fig. 6, we examine the Diversity-Fidelity Trade-off by fixing exploration and regularization levels instead of using dynamic $p_t$ for balancing. Results show that stronger regularization undermines preference performance without enhancing diversity, while higher exploration reduces diversity through image distortion. In contrast, our method consistently surpasses baselines and achieves a balanced trade-off between richness and alignment.

## 4 CONCLUSION

We propose *LitExplorer*, a training-free plugin framework for diffusion models. It integrates an Inheritance–Restart exploration mechanism and a Quality–Efficiency arbitration framework to prevent early convergence, enhance high-reward exploration, and prune irrelevant signals. Experiments show consistent gains in quality, diversity, and efficiency over existing methods.

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

APPENDIX

# A RELATED WORK

## A.1 DIFFUSION MODEL

Diffusion models have achieved remarkable progress in generative modeling. Their stability and gradual refinement allow them to approximate complex distributions, producing samples with both high fidelity and diversity. They have shown strong generalization across domains, including text-to-image (Li et al., 2024; Yuan et al., 2024; Ding et al., 2025), speech (Kong et al., 2020; Chen et al., 2020; Liu et al., 2022), molecular design (Xu et al., 2022; 2023b), and protein generation (Watson et al., 2023; Abramson et al., 2024; Qiao et al., 2022). However, since these models are typically pretrained on broad datasets lacking task-specific signals, they often fail to reflect user-desired distributions in specialized applications. This gap highlights the need for alignment:enabling customized optimization toward target distributions without losing generality.

## A.2 GRADIENT-BASED FINE-TUNING METHODS

*Direct gradient-based fine-tuning is intuitively feasible for enhancing diffusion models. For examples, DRaFT Clark et al. (2023) performs differentiable reward optimization for diffusion models by backpropagating reward gradients through sampling steps, forming a unified gradient-based framework for preference-aligned fine-tuning; DRTune Wu et al. (2024) stabilizes reward-supervised diffusion training by applying stop-gradient to denoiser inputs and uniformly sampling K timesteps, enabling efficient supervision across all sampling depths; Diffusion-RPO Gu et al. (2024) extends relative preference optimization to diffusion models by conducting stepwise preference contrasts with CLIP-based multimodal weighting to learn coherent cross-semantic preference patterns. All these methods improve preference alignment with limited stability, scalability, and robustness under diverse real-world undifferentiated reward signals.*

## A.3 REINFORCEMENT LEARNING FINE-TUNING METHODS

Existing alignment approaches fall into two categories: fine-tuning (Black et al., 2023; Fan et al., 2023) and training-free methods. The mainstream fine-tuning approach relies on reinforcement learning (RL) (Sutton, 2018) to adjust diffusion model weights. RL greatly improves alignment to downstream targets but faces challenges: rewards are only available after the full denoising process, making them sparse and prone to reward hacking. This often leads to higher reward scores but reduced diversity and poor distribution coverage. Moreover, RL-based fine-tuning comes with a high computational cost (Kim et al., 2025).

## A.4 TRAING-FREE METHODS

Training-free methods (Kim et al., 2025; Xie & Gong, 2025) provide an alternative: instead of changing diffusion model weights, they bias the sampling process at inference to guide results toward target objectives. *For examples, TTScale Ma et al. (2025) boosts generation quality by shifting extra inference compute from longer denoising chains to noise-search guided by a lightweight evaluator. FKSteer Singhal et al. (2025) enables inference-time controllability by using particle-based weighting and resampling to steer diffusion trajectories without fine-tuning.* This avoids costly retraining and reduces computation. First, prior work lacks an adaptive mechanism to balance diversity and fidelity (Dhariwal & Nichol, 2021). Second, existing training-free methods often overemphasize target guidance, forcing trajectories into narrow regions and reducing coverage of the pretrained distribution. While less prone to severe reward hacking, this effect still harms diversity. Further, guidance signals are typically applied indiscriminately across all steps (Tang et al., 2024), without filtering ineffective or saturated signals. This overuse increases inference cost and amplifies noise, sometimes even causing negative guidance effects.

# B  METHOD SUPPLEMENT

## B.1  CONTROL NETWORK

We assign $y = 0$ to samples close to pure noise $\mathbf{x}_T$, and $y = 1$ to samples close to the target $\mathbf{x}_0$, pretraining the network via the loss defined as follows.

$$\mathcal{L}(\theta) = -\mathbb{E}_{(\mathbf{x}_t, y)} \left[ y \log f_\theta(\mathbf{x}_t) + (1 - y) \log(1 - f_\theta(\mathbf{x}_t)) \right]. \tag{10}$$

# C  ADDITIONAL EXPERIMENT RESULTS

## C.1  DETAILED HYPER-PARAMETER SETTING

Table 5: Hyper-parameters and metrics in our experiment. *Notably, only $n$ and $\delta_r$ are newly introduced by our method; all other hyper-parameters follow DDIM or the baseline methods.*

| Name | Description | Value |
|------|-------------|-------|
| $\eta$ | eta parameter for the DDIM sampler | 1.0 |
| $w$ | classifier-free guidance weight | 5.0 |
| $m_{dno}$ | Iteration Number for DNO | 20 |
| $m_{dymo}$ | Iteration Number for DyMO | adaptive |
| $n_{das}$ | Particle number for DAS | 4 |
| $mp$ | mixed precision | fp16 |
| $n$ (LitExplorer-*unique*) | Exploration supplement number | 2 |
| $\delta_r$ (LitExplorer-*unique*) | Reward enhancement threshold | 0.1 |

The full list of hyper-parameters in our paper is shown in Table 5. Then, we provide a detailed illustration of each introduced metric:

- **Aesthetic Preference**
  - **AES (Aesthetic Score)** AES is a metric used for evaluating the aesthetic quality of an image, typically by training a model to predict human aesthetic ratings (Schuhmann et al., 2022).
  - **PS (PickScore)** PS is a method based on human selection preferences for rating the aesthetic quality of images (Kirstain et al., 2023).
  - **IR (ImageReward)** IR is a metric used for assessing image quality, often in the optimization process of image generation models (Xu et al., 2023a).
  - **HPSv2 (Human Preference Score v2)** HPSv2 is a preference-based metric that predicts human preferences for generated images. This model fine-tunes the CLIP model on the HPD v2 dataset. HPSv2 excels in multiple styles, including animation, concept art, paintings, and photographs (Wu et al., 2023).

- **Image Fidelity**
  - **ClipScore** ClipScore is a model-based image scoring metric that evaluates the similarity between a generated image and its textual description. It calculates the cosine similarity between the image and text embeddings. The metric is highly correlated with human judgment and does not require reference text (Radford et al.).
  - **Fréchet Inception Distance (FID)** FID is a metric used to evaluate the quality of generated images. It compares the mean and covariance of the features extracted from real and generated images using the Inception v3 model. A lower FID value indicates that the generated image is closer to real images (Heusel et al., 2017).
  - **Improved F1 Score (iFS)** iFS is an improvement over the traditional F1 score, aiming to provide a better evaluation of generative model performance. It optimizes the balance between precision and recall for more accurate assessments (Kynkäänniemi et al., 2019).

- **Generative Diversity**

Table 6: Results with SDv14.

| Method | IR | PS | CLIP | LPIPS | TCE | NIQE↓ |
|---|---|---|---|---|---|---|
| SDv14 | 0.744 | 20.62 | 0.240 | 0.671 | 40.32 | 4.6312 |
| DDPO | 1.056 | 21.20 | 0.240 | 0.609 | 39.61 | 4.554 |
| DNO | 0.915 | 21.85 | 0.239 | 0.597 | 39.24 | 4.875 |
| DAS | 1.217 | 22.58 | 0.238 | 0.610 | 39.56 | 5.451 |
| DyMO | 1.100 | 23.14 | 0.235 | 0.615 | 39.63 | 4.550 |
| Ours | 1.230 | 23.39 | 0.242 | 0.676 | 41.10 | 3.494 |

Table 7: Results with SDv21.

| Method | IR | PS | CLIP | LPIPS | TCE | NIQE↓ |
|---|---|---|---|---|---|---|
| SDv21 | 0.541 | 20.63 | 0.280 | 0.644 | 39.69 | 5.672 |
| DDPO | 0.794 | 21.25 | 0.274 | 0.622 | 38.94 | 4.953 |
| DNO | 0.566 | 20.91 | 0.277 | 0.589 | 39.13 | 5.551 |
| DAS | 0.817 | 22.29 | 0.280 | 0.656 | 39.55 | 5.150 |
| DyMO | 0.921 | 22.62 | 0.279 | 0.623 | 39.62 | 5.176 |
| Ours | 0.980 | 22.91 | 0.280 | 0.677 | 40.07 | 3.695 |

- **LPIPS (Learned Perceptual Image Patch Similarity)** LPIPS is a metric for evaluating perceptual similarity between images. It compares the feature activations of images in pre-trained convolutional neural networks (e.g., VGG). A higher LPIPS value indicates lower perceptual similarity and a higher diversity(Zhang et al., 2018).

- **TCE (Truncated CLIP Entropy)** Truncated CLIP Entropy (TCE) is a measure of semantic diversity of a set of generated images, computed in the joint image–text embedding space of CLIP. It is defined by first mapping each image $x_i$ to a vector $\mathbf{z}_i = \text{CLIP}_{\text{img}}(x_i) \in \mathbb{R}^d$, then forming the empirical covariance matrix $\Sigma$ of these vectors, extracting its top-$k$ eigenvalues $\lambda_1 \geq \cdots \geq \lambda_k$, and finally computing

$$\text{TCE}_k = \tfrac{1}{2} \sum_{i=1}^{k} \log \lambda_i.$$

  TCE gives a tractable proxy for how "spread out" the images are in the CLIP semantic space. (Ibarrola & Grace, 2024).

- **Inception Score (IS)** IS is a metric used to evaluate the quality of generated images. It calculates the entropy of the class distribution of images in a pre-trained Inception v3 model. Higher IS values indicate that the generated images are both clear and diverse (Salimans et al., 2016).

- **Compositional Richness**

  - **NIQE (Natural Image Quality Evaluator)** NIQE is a no-reference image quality assessment metric. It evaluates the image quality by measuring the distance between the natural scene statistics of the image and a natural image database of undistorted images (Mittal et al., 2012b). A lower NIQE value indicates better image quality.

  - **BRISQUE (Blind/Referenceless Image Spatial Quality Evaluator)** BRISQUE is another no-reference image quality metric. It evaluates image quality by analyzing spatial-domain features. It does not rely on reference images and is suitable for a variety of image quality assessment tasks (Mittal et al., 2012a).

  - **Spectral Entropy (SE)** SE is a metric for evaluating the spectral properties of images. It calculates the entropy value of an image's frequency spectrum to assess the complexity and richness of image details. Higher SE values generally indicate more detailed and textured images (Liu et al., 2014).

Table 8: Results of Restart Count and Early Stop across HPSv2 and Pick-a-Pic with SDv15 and SD-XL backbones.

| Avg. Value | Base | HPSv2 | | Pick-a-Pic | |
|---|---|---|---|---|---|
| | | SDv15 | SD-XL | SDv15 | SD-XL |
| Restart Count | 50 | 4.37 | 8.55 | 7.32 | 10.59 |
| Guiding Step | 50 | 37.35 | 44.21 | 40.67 | 44.30 |

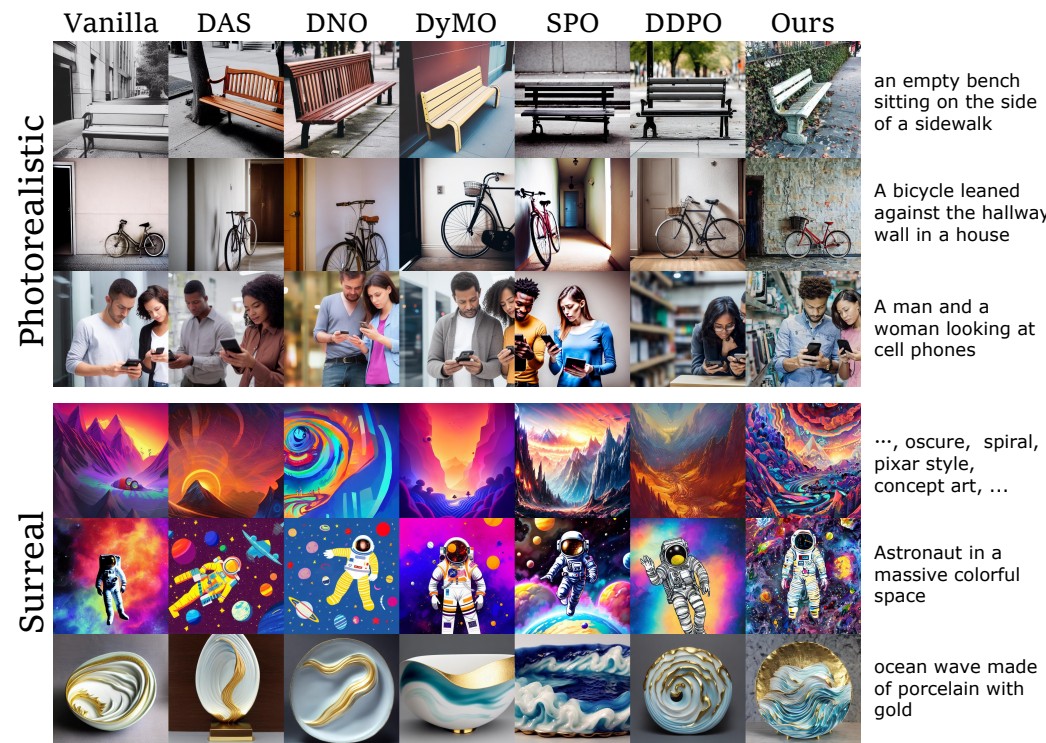

Figure 7: The visual comparisons of different methods on SDv15. For photorealistic prompts from HPSv2, our results contain richer details while maintaining image-prompt alignment. For surreal prompts from Pick-a-Pic, we exhibit much more diverse patterns and colors.

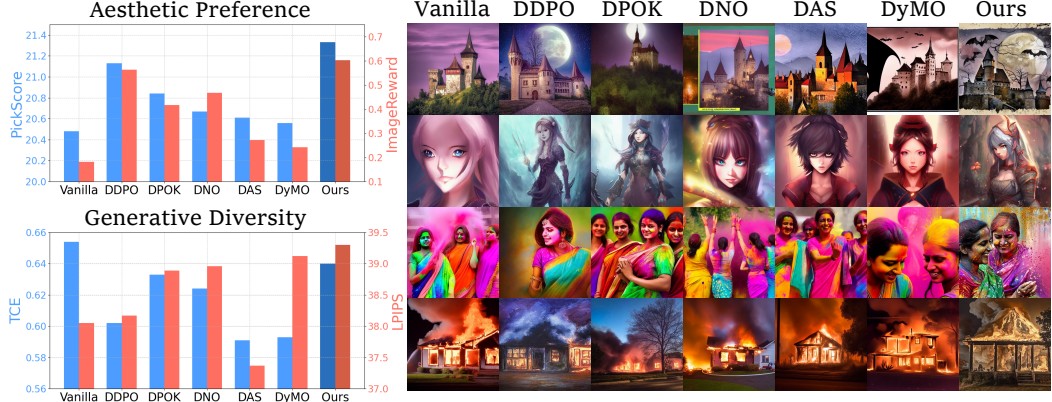

Figure 8: Results with Aesthetic Score as the optimization objective.

## C.2 SUPPLEMENTARY STATISTICAL SIGNIFICANCE

*Since the generative quality is highly related to the initial noise decided by the seeds, we further conduct experiments with 5 different seeds and calculate their average Coefficient of Variation (CoV). The metrics and evaluation protocol are strictly following Tab. 1. As shown in Tab. 9, it can be observed that all methods exhibit similarly low CoV across various seeds, which indicates that the guidance-based training-free scaling methods perform relatively stably. These results further demonstrate that the performance reported in Table 1 is statistically significant.*

Table 9: *CoV (%) comparison of methods across four evaluation metrics.*

| Method | Preference | Fidelity | Diversity | Richness |
|--------|-----------|----------|-----------|----------|
| DNO | 0.88 | 0.52 | 0.27 | 1.27 |
| DAS | 0.96 | 0.81 | 0.30 | 1.36 |
| DyMO | 0.62 | 0.86 | 0.28 | 1.15 |
| Ours | 0.65 | 0.79 | 0.28 | 1.41 |

Table 10: *Mean step numbers predicted by the control network across different datasets. The control network is trained on the simple animal and directly generalizes to other datasets.*

| Method | Total | SimpleAnimal | Pick-a-Pic | HPSv2 | GenEval |
|--------|-------|--------------|------------|-------|---------|
| SD15 | 50 | 37.3 | 40.9 | 39.8 | 38.1 |
| SDXL | 30 | 24.7 | 24.9 | 25.1 | 24.6 |

## C.3 GENERALIZABILITY OF PRE-TRAINED CONTROL NETWORK

*Considering training a control network could undermine the claim of "training-free" of the proposed LitExplorer, it is necessary to demonstrate that the Control Network could also be seen as a pre-trained scorer, which can be pre-trained once and be generalizable for all. Therefore, as shown in Tab. 10, we pre-train the control network on the simple animal set and predict steps on other datasets. The results demonstrate that the network could easily generalize to other datasets with similar effective performance, which is caused by the fixed scheduler having already pre-set the noise level for different steps in a coarse manner.*

## C.4 ANALYSIS ON PERFORMANCE-COMPUTATION TRADE-OFF

*Due to the iterations or sampling numbers can influence the performance of inference-time scaling methods, we investigate the relation between computational cost and performance. Specifically, we first deploy extra mean number of Function Evaluated (nFE) to quantify the computational cost in a unified manner, which represents the number of extra iterations or sampling in one denoising step. Then, we compare our method with iteration-based DyMO and sampling-based DAS, where the maximum iteration and sampling particle number are set to 2,4,8,16. Notably, LitExplorer includes both sampling and iteration, we hence decouple these factors and design two variants of our method, that is, Ours-i (iteration) and Ours-p (particles). As shown in Fig. 9, our results can firstly save nFE because we conduct early stopping. Second, the curves of our results consistently surpass the baselines, indicating the consistent superiority of LitExplorer.*

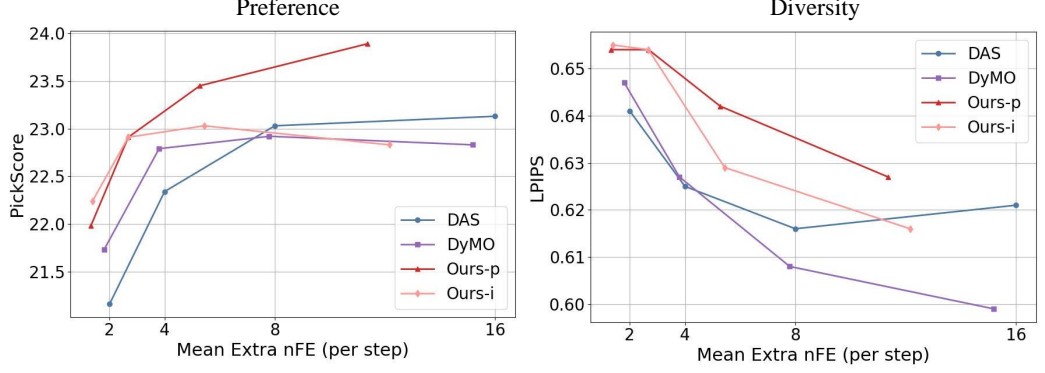

Figure 9: *Investigation on Performance-Computation Trade-off.*

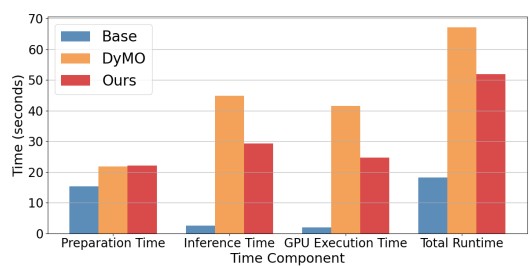

Figure 10: *Wall-clock runtime comparisons with details.*

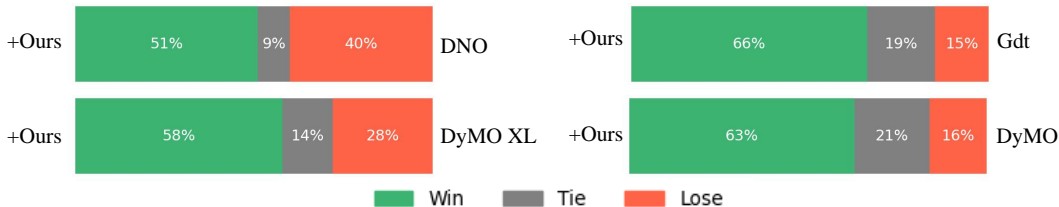

Figure 11: *User study for plug-and-play effectiveness of our method.*

## C.5 COMPARISONS ON WALL-CLOCK RUNTIME.

*To investigate the effectiveness in real-world application, we conduct comparisons on wall-clock runtime with details. As shown in Fig. 10, we present the time consumption of running the entire script. Firstly, their preparation time are similar and once prepared, the script can conduct arbitrary times of inference. Then, it can be observed that our inference time is much faster than the baseline DyMO, while Base has the smallest inference time since it includes no optimization or guidance.*

## C.6 USER STUDY

*To subjectively evaluate the generative quality of different methods, we conduct a user study based on the images from the Plug-and-Play experiments in Sec. 3.3. Specifically, we recruit five subjects without prior knowledge to select the preferred image within paired images with the corresponding prompt as the reference. As shown in Fig. 11, our method consistently performs as an effective plugin that achieves superior quality compared to the implemented original models.*

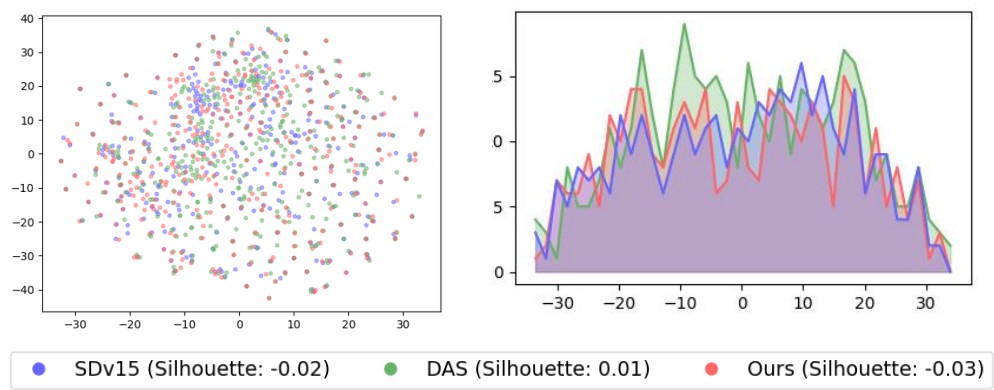

Figure 12: *Feature visualization comparison with DAS.*

## C.7 SUPPLEMENTARY DISTRIBUTION COMPARISON WITH DAS

*In Fig. 2, we compare the feature distribution with the iteration-based DyMO, thus demonstrating the diversity of our method. Here, we conduct a similar experiment to evaluate the sampling-based baseline, i.e., DAS. As shown in Fig. 12, although the sampling-based DAS has richer diversity compared to DyMO, it still tightens the distribution range of the original backbone. In contrast, our result exhibits consistent diversity, which is demonstrated by the wider visualized distribution and the lower Silhouette Coefficient value.*

Table 11: *Comparison results with the backbone of SD3. Gdt represents the baseline that directly applies the reward gradient to the latent code during inference. Ours is designed based on Gdt with the proposed plugin components.*

| Method | PS | CLIP | LPIPS | NIQE↓ |
|--------|-------|-------|-------|-------|
| SD3 | 21.62 | 0.238 | 0.645 | 4.312 |
| Gdt | 22.17 | 0.239 | 0.636 | 4.304 |
| Ours | 22.57 | 0.241 | 0.641 | 2.990 |

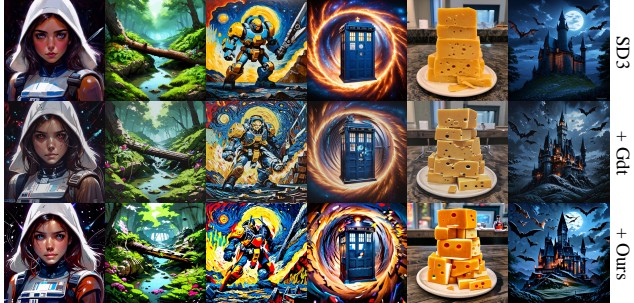

Figure 13: *Visual impression on SD3. In comparison, our results exhibit finer details, more diverse colors, and improved aesthetic style.*

## C.8 SUPPLEMENTARY RESULTS ON DiT-BASED ADVANCED BACKBONE

*To further validate the effectiveness of LitExplorer, we implement it to the advanced Stable Diffuion V3 backbone with PS as the reward and Pick-a-pic as dataset. Quantitatively, as shown in Tab. 11, our method can still effectively improve the reward score while maintaining the alignment and diversity. Qualitatively, it can be clearly observed that our method has achieved improved human preference in aesthetics.*

## C.9 VISUAL QUALITY

To evaluate the generative quality, we compare some sampling results in Fig. 7, which further demonstrates the superiority of our method. Moreover, as shown in 8, the results of guiding via aesthetic score indicate our overall superiority on aesthetic preference and generative diversity. Then, in the visual results, it could be clearly observed that our generated images contain richer details, better alignment, and improved aesthetic impressions.

## C.10 ANALYSIS ON EXPLORATION.

To demonstrate the exploration effect of introducing ESV, we present the intermediate results during generation. As shown in Fig. 14, our method exhibits superior exploration effect during denoising, implying the improved generative quality.

## C.11 FURTHER ANALYSIS ON ABLATION STUDY

*The proposed LitExplorer is designed considering two trade-offs, that is, Diversity-Fidelity and Efficiency-Quality trade-offs. Therefore, different components are introduced with paired correlations. Specifically, introducing M-ESV can introduce exploration and thus improve diversity, while L2 can maintain the fidelity to prevent over-exploration that leads to the collapse of the denoising process. Therefore, without ESV that encourages exploration, solely deploying L2 is unnecessary. Then, both I&R and EarS are introduced to enhance the generation quality based on the exploration. Meanwhile, GuiS builds upon the previously introduced exploration, as direct guidance typically provides a straightforward path to higher rewards, making GuiS ineffective in such cases.*

A girl holding a glowing teddy bear lantern, surrounded by softly falling snow, ...

A man standing under cherry blossoms in full bloom, ⋯, cinematic storytelling portrait

Figure 14: The proposed ESV facilitates exploration during the denoising process. **Left**: ESV enables our method to identify a broader range of patterns and adjust the trajectory toward the "teddy bear lantern" during intermediate stages. **Right**: Our method exhibits greater posture variation throughout denoising. While this variation is not directly responsible for the superior final generation, it reflects an increased level of exploration.

*In contrast, with ESV, guidance is not based on the current state but rather incorporates an exploratory shift, allowing GuiS to effectively filter out harmful optimization. As for the intensity of different components in the trade-offs, we present Fig. 6 to demonstrate the superiority of our adaptive strategy. Overall, the components of our approach are interdependent and mutually reinforcing, collectively contributing to the superior performance of our image generation scheme in terms of inference-time scaling.*

## C.12 RESULTS ON MORE SD BACKBONES.

In addition to SDv15 and SD-XL presented in the main paper, we provide the results on SDv14 and SDv21-turbo. As shown in Tab. 6 and 7, our method also leads in all metrics.

## C.13 EXHIBITED ADAPTIVE VALUES FOR COMPUTATION REDUCTION.

Here, we provide the exhibited average values of restart counts and early stop positions. Specifically, we calculate the average restart counts and average number of guiding steps on HPSv2 and Pick-a-Pic datasets with SDv15 and SD-XL. As shown in Tab. 8, it can be observed that HPSv2 has smaller values of both variables compared to Pick-a-Pic, while SDv15 has smaller values than SD-XL. This is because HPSv2 has more photorealistic images that are easier than surreal one for denoising, while SDv15's lesser robustness makes it more amenable to external guidance.

## D DECLARATION OF LLM USAGES.

While we utilized an LLM to assist in polishing the English for improved clarity, all aspects of idea development, theoretical validation, and experiments were carried out solely by the authors, without LLM interference.

### D.1 SHOWCASE PROMPT TABLE

Please refers to Tab. 12, 13, 14, and 15.

Table 12: Detailed prompts used for generated images in Fig. 1

| Image | Prompt |
|---|---|
| Fig. 1, Row 1, Col 1 | ballet dancer, insanely detailed, photorealistic, 8k, perfect composition, volumetric lighting, natural complexion, award-winning professional photography, taken with Canon EOS 5D Mark IV, 85mm, mindblowing, masterpiece |
| Fig. 1, Row 1, Col 2 | A fluffy bunny as Rapunzel, with long golden ears flowing down from a tall enchanted tower, glowing lanterns in the night sky, warm fairytale atmosphere |
| Fig. 1, Row 1, Col 3 | A dog in sportswear lifting tiny dumbbells at the gym, determined expression, humorous fitness illustration |
| Fig. 1, Row 1, Col 4 | A boy superhero landing on the ground in classic "hero pose," debris and glowing sparks flying around, comic action shot |
| Fig. 1, Row 1, Col 5 | A phoenix rising up from ashes |
| Fig. 1, Row 2, Col 1 | A young girl standing on a rooftop, blowing dandelions that transform into glowing comets, shooting across the night sky, dreamy fantasy artwork |
| Fig. 1, Row 2, Col 2 | A small hedgehog as the Frog Prince, wearing a tiny crown while sitting on a lilypad, a kind-hearted swan princess leaning close, surrounded by glowing fireflies, magical fairytale illustration |
| Fig. 1, Row 2, Col 3 | a white polar bear cub wearing sunglasses sits in a meadow with flowers. |
| Fig. 1, Row 2, Col 4 | A cat surfing on a giant wave at sunset, wearing cool shades, cinematic sports illustration |
| Fig. 1, Row 2, Col 5 | A sunflower in full bloom under golden sunlight, tiny dewdrops sparkling on its petals, cinematic macro fantasy illustration |
| Fig. 1, Row 3, Col 1 | A warrior standing at the edge of a glowing crater, surrounded by swirling cosmic energy, their silhouette outlined against the birth of a new star, ultimate epic fantasy art |
| Fig. 1, Row 3, Col 2 | A boy lying on the grass in a field, listening to music with glowing headphones, fireflies surrounding him |
| Fig. 1, Row 3, Col 3 | A little girl painting a rainbow bridge from the classroom window into the sky, playful magical fairytale art, hopeful and inspiring |
| Fig. 1, Row 3, Col 4 | A group of playful penguins throwing glowing snowballs at each other, each snowball turning into sparkling stars when it explodes, magical fairytale scene |
| Fig. 1, Row 3, Col 5 | Giant rubber duck floating in the ocean with a small island on its back, surrounded by tropical palm trees and crystal clear water, bright and sunny day, calm seas, vivid colors, cinematic lighting, high detail |
| Fig. 1, Row 3, Col 6 | A cozy library built inside an ancient oak tree, warm lights glowing through round windows, whimsical fairytale healing atmosphere |
| Fig. 1, Row 3, Col 7 | A brave boy carrying a glowing lantern, releasing trails of light that form a golden sunrise, cinematic epic fantasy style |

Table 13: Detailed prompts used for generated images in Fig. 8

| Image | Prompt |
|---|---|
| Fig. 8, Row 1 | transylvania castle on hiltop and dusk bats and scifi |
| Fig. 8, Row 2 | fantasy character portrait digital painting, anime style, detailed with beautiful emotive lighting suggesting personality, and background that suggests character backstory |
| Fig. 8, Row 3 | Women in Saree playing Holi |
| Fig. 8, Row 4 | a house burning at night |

Table 14: Detailed prompts used for generated images in Fig. 4

| Image | Prompt |
|---|---|
| Fig. 4, Row 1 | a slice of pizza floating through space with stars in the background |
| Fig. 4, Row 2 | A Great Dane dog in the style of Vincent Van Gogh |

Table 15: Detailed prompts used for generated images in Fig. 6

| Image | Prompt |
|---|---|
| Fig. 6, Row 1 | Heart shaped balloon |
| Fig. 6, Row 2 | An apple on a table |
| Fig. 6, Row 3 | A castle in the sky, clouds, sunset, explosion |
| Fig. 6, Row 4 | Harry potter as a cat, pixar style, octane render, HD, high-detail |

