# OpenReview forum: "LitExplorer: Training-Free Diffusion Guidance with Adaptive Exploration-Filtering Framework"
_ICLR.cc/2026/Conference — Submitted to ICLR 2026_

### Official Review · Reviewer_6nmv · 2025-11-01

**Soundness:** 2
**Presentation:** 2
**Contribution:** 2
**Rating:** 4
**Confidence:** 4

**Summary:**

This paper focuses on training-free methods for aligning diffusion models with specific objectives while maintaining diversity, fidelity, and efficiency. The proposed method adopts an exploration technique to control the diversity-fidelity tradeoff and adaptive guidance to control the quality-efficiency tradeoff. Experiments are conducted with image generation with various objectives to compare with prior works.

**Strengths:**

1. The paper clearly identifies the problem of diversity degradation after alignment and the inefficiency of previous guidance methods.
2. Multiple metrics are incorporated to holistically evaluate preference, fidelity, diversity, and richness.
3. The proposed method improves preference, richness, and speed over the base model without sacrificing diversity or fidelity.

**Weaknesses:**

1. The method is mostly based on heuristics, and the design choices are not clearly justified. Though they provide ablation on diversity and diversity-fidelity tradeoff, it doesn't assess each component of the methods, thus their claims regarding each component lack evidence. (See Question 1, 2)
2. Quantitative results presented in Section 3 don't contain margin of error, hence it's hard to assess how much the proposed method outperforms prior methods.
3. The method neads to train a Control Network, so it's not totally training-free.

**Questions:**

1. How does exploration supplement variable help increase diversity? If the selection from the candidates is done randomly, it can increase diversity. However, the method uses a reward to select the highest reward sample, which actually decreases diversity as the number of candidates increases by always selecting the optimal candidate.
2. The motivation of 'inheritance' in inheritance-restart is unclear. The noise level, i.e., the magnitude of the exploration supplement variable, should be kept along with denoising to keep the sample on-manifold. Also, why would reusing the previous exploration supplement variable help improve?

---

> ### Author Response · Authors · 2025-11-24
> **Point-by-Point Response to the Esteemed Reviewer 6nmv (part 1)**
>
> Thank you for your review. We are truly honored by your recognition of our motivation and the extensive experimental evaluation. Regarding your questions and suggestions, we have carefully considered them and made the corresponding additions and modifications in the newly uploaded PDF (revisions marked by italy blue). Below, we provide a point-by-point detailed response to your feedback.
>
> ---
>
> # Weakness 1
> > The method is mostly based on heuristics, ... .(See Question 1, 2)
>
> Thank you for your valuable suggestion. For a detailed response to this issue, please refer to our replies to **Q1** and **Q2**.
>
> ---
>
> # Weakness 2
> > Quantitative results presented in Section 3 don't contain margin of error, hence it's hard to assess how much the proposed method outperforms prior methods.
>
> First, our results in Tab. 1,2,3 are based on testing with the same set of **500 prompts from Pick-a-pic or HPSv2** and the same seeds across all the compared methods. The reported results are the **averages of various metrics** across all images, which already represent a **statistical measure with fair and consistent settings**, demonstrating the effectiveness of our method.
>
> Additionally, we calculated the **coefficient of variation (CoV)** for the mean of different metrics across five seeds for the training-free baselines. These results are presented in **Appendix Tab. 9**. As shown, the CoV for the mean scores of each method does not exhibit significant relative differences, and the absolute values are relatively small. This is likely because these inference-time methods involve **sample-wise optimization**, which results in similar adaptability for each sample, preventing generalization issues that could arise from deviations from the training distribution.
>
> In conclusion, the comprehensive experimental results provide strong evidence that our method demonstrates significant superiority.
>
> ---
>
> # Weakness 3
> > The method neads to train a Control Network, so it's not totally training-free.
>
> The **Control Network** we use can be viewed as a **pretrained** network model, and the process through which it is obtained is **decoupled and independent** from the method we propose. Given that the **scheduler** generally controls the denoising intermediate state variations for all prompts in a relatively consistent manner, this model exhibits very good generalization. In other words, it can be **pre-trained once and for all**, and therefore should not be counted as part of the training process.
>
> Similarly, the **pretrained scorer**, such as **PickScore**, is also a model trained at scale beforehand. It would not be correct to argue that using **PickScore** disqualifies our approach from being considered training-free. Just as **PickScore** is used to score images after pretraining, the **Control Network** can be viewed as a simple pretrained scoring model for noise.
>
> **Experimentally**, we evaluated the generalization of the Control Network in the appendix **Tab. 10**, demonstrating that it can be effectively **pre-trained once and for all**, and therefore should not be considered as part of the training process but rather as a readily available pretrained model (like PickScore).
>
> In conclusion, our method is indeed **training-free**.
>
> ---
>
> # Question 1
> > How does exploration supplement variable help increase diversity? If the selection from the candidates is done randomly, it can increase diversity. However, the method uses a reward to select the highest reward sample, which actually decreases diversity as the number of candidates increases by always selecting the optimal candidate
>
> First, from a **theoretical perspective**, the original distribution of the model can be considered as $p_0(x)$. When we further introduce exploration $\alpha \in \mathcal{N}(0, I)^N$, we are essentially augmenting the distribution to $p\_{\alpha}^r(x\_0)$. Then, we select the high-reward one among multiple choices.
>  That is,
> - Introducing reward guidance itself is a mechanism that reduces diversity.
> - The first step contributes to increasing diversity.
>
> In summary, we **enhance diversity under the constraint of reward guidance**, and our method increases the distribution range more than other inference-time scaling methods, while attempting to maintain the diversity of the original, un-guided model.
>
> **Experimentally**, as shown in **Fig. 2, 12**, our method exhibits a **wider distribution range** compared to the guided baseline, and it is closer to the **original model**. The results in **Tab. 1, 2, 3** further highlight that while the original, un-guided model exhibits the best diversity, its human preference quality is lower. On the other hand, existing guided baselines significantly lose diversity, whereas our method enhances **human preferences** and maintains **the diversity of the original model**.

---

> ### Author Response · Authors · 2025-11-24
> **Point-by-Point Response to the Esteemed Reviewer 6nmv (part 2)**
>
> # Question 2
> > The motivation of 'inheritance' in inheritance-restart is unclear. The noise level, i.e., the magnitude of the exploration supplement variable, should be kept along with denoising to keep the sample on-manifold. Also, why would reusing the previous exploration supplement variable help improve?
>
> Thank you for your comment.
>
>
> ### 1. **Motivation Behind 'Inheritance'**
> The core motivation behind our proposed **'inheritance'** method is to save computational resources, as highlighted by the title of our work, **LitExplorer**.
> Simply put, **inheritance** can be seen as reusing the optimization results of the **exploration supplement variable** from the previous step as the initialization for the current step.
> By doing this, we introduce exploration at each step while minimizing the computational overhead required for optimization, leading to improved speed.
>
> ### 2. **Role of Inheritance-Restart**
> The **inheritance-restart** mechanism is introduced to handle situations where the previous step’s variable is not suitable for the current step.
> When this happens, **inheritance-restart** restarts the random search. This enhances randomness while, together with the **L2 constraint** in **Eq. (8)**, ensuring that the process stays **on-manifold**, which stabilizes the inference process.
>
> ### 3. **Experimental Impact**
> Experimentally, we can observe the combined impact of **inheritance-restart** on both **time** and **quality**.
> As shown in **Tab. 4**, please focus on **V3** and **V4**.
> It can be observed that introducing **inheritance-restart** effectively reduces time consumption while ensuring that the optimization performance is maintained.

---

### Official Review · Reviewer_QPUb · 2025-11-03

**Soundness:** 2
**Presentation:** 2
**Contribution:** 2
**Rating:** 4
**Confidence:** 3

**Summary:**

The paper introduces a test-time alignment framework for text-to-image diffusion. The method aims to balance exploration and refinement while also improving efficiency during sampling. To achieve this, the approach combines reward-guided candidate sampling, a progression-aware control network, and screening and early-stop criteria to avoid unnecessary guidance. Experiments on SD v1.5 and SDXL with multiple reward models (PickScore, AES, ImageReward) show consistent improvements in reward-aligned image quality without requiring model fine-tuning.

**Strengths:**

1.	The framework is conceptually clear, separating exploration–exploitation and quality–efficiency in a straightforward manner.
2.	The method operates in a plug-and-play manner with respect to the diffusion backbone. This makes it easy to plug into existing pipelines without T2I diffusion model updates.
3.	The experiments shows consistent improvements across multiple reward signals.

**Weaknesses:**

1. While the paper emphasizes being "training-free", the framework relies on a learned network to estimate how close the current state is to the clean data sample. However, the training setup, dataset scale, and generalization capability of this network are not described in sufficient detail, and there is no widely recognized pretrained model that can be adopted for this role. This dependency weakens the claim that the method is truly "training-free", making it closer to a "test-time plugin" that still requires a separate trained module.

2. The paper presents a compelling conceptual framing through expolration-exploitation and quality-efficiency balancing, but the actual pipeline consists of multiple components and eight hyper-parameters. This makes the method less modular than suggested. Specifically, Table 4 does not fully reveal the isloated impact of eac hmodule, and the hyper-parameters settings in Table 5 lack sufficient justification for their chosen values. A more thorough study on component interactions and hyper-parameter sensitivity would strengthen the work, particularly since reward-guided generation is known to be sensitive to reward scale and optimization dynamics [1].

3. The paper does not compare against recent fine-tuning based alignment approaches such as [2] DRaFT, [3] Diffusion-RPO, and [4] DRTune. Adding these baselines would better contextualize the claimed advantages in quality and efficiency compared to fine-tuning based method.

(Minor)
1. The experiments do not present an execution time (or a wall-clock runtime) comparison on overhead breakdown for the full pipeline.

2. The experiments primarily focus on reward-model metrics without human preference studies or user evaluation. While these scores improve, it remains unclear whether the gains reflect real perceptual quality rather than reward-model alignment. Incorportaing even a small-scale human evaluation would make the empirical validation better.

3. The paper shows experimental results on U-Net based models(SDv15 and SD XL1.0), and it would be better to explore its applicability to recent Transformer-based models such as FLUX or SD3.

**Questions:**

Q1. Diffusion sampling assumes that intermediate states lie approximately on a perturbed noise manifold with respect to the forward process [5], [6]. Since the method introduces additional gradients and regularization signals during sampling, could these guidance terms push intermediate latents off the diffusion trajectory? How does the regularizations work within this method in terms of preverving manifold during denoising process.
Q2. [Typos] $x^{*}$ in Line 226 and $x^{⋆}$ in Equation (7) are not clealy distinguished. Also, $p_t$ in line 299 and $\alpha_t$ in  Figure 3 should be unified.

[1] Subject-driven Text-to-Image Generation via Preference-based Reinforcement Learning
[2] Directly Fine-Tuning Diffusion Models on Differentiable Rewards
[3] Diffusion-RPO: Aligning Diffusion Models through Relative Preference Optimization
[4] Deep Reward Supervisions for Tuning Text-to-Image Diffusion Models
[5] Diffusion Posterior Sampling for General Noisy Inverse Problems
[6] Manifold Preserving Guided Diffusion

---

> ### Author Response · Authors · 2025-11-24
> **Point-by-Point Response to the Esteemed Reviewer QPUb (part 1)**
>
> Thank you for your careful review and insightful comments. We are encouraged by your recognition of the concepts we have introduced and the breadth of our experimental evaluation. In response to your feedback, we have thoroughly revised the paper and uploaded a new PDF, with the changes highlighted in **blue italics**. Next, we will provide a point-by-point response to your questions, aiming to meet your high standards.
>
> ---
>
> # Weakness 1
> > While the paper emphasizes being "training-free", the framework relies on a learned network to estimate how close the current state is to the clean data sample. However, the training setup, dataset scale, and generalization capability of this network are not described in sufficient detail, and there is no widely recognized pretrained model that can be adopted for this role. This dependency weakens the claim that the method is truly "training-free", making it closer to a "test-time plugin" that still requires a separate trained module.
>
> Thank you very much for your insightful question. You raise a valid point, which demonstrates your mature thinking and relevant knowledge. In fact, we had already considered this issue. Here is our explanation as to why our method can be considered **training-free**:
>
> ### 1. **Control Network as a Pretrained Model**
> The **Control Network** can indeed be viewed as a pretrained model because:
>
> - **Independence from our method**: The process of obtaining the **Control Network** is decoupled and independent from the method we propose. It does not require separate training at inference time for each use.
> - **Good generalization**: Given that the **DDIM scheduler** controls the denoising intermediate states with relatively consistent noise across all prompts, the model demonstrates **excellent generalization**. This means it can **pre-train once and for all**, similar to how **PickScore** operates.
>
> ### 2. **Experimental Validation**
> Experimentally, we evaluate the **generalization of the Control Network** in **Appendix Table 10**. The results demonstrate that it can be effectively **pre-trained once and for all**, and therefore should **not** be counted as part of the training process. Instead, it should be treated as a readily available pretrained model (just like PickScore).
>
> In summary, our method is indeed **training-free**.
>
> # Weakness 2
> > The paper presents a compelling conceptual framing through expolration-exploitation and quality-efficiency balancing, but the actual pipeline consists of multiple components and eight hyper-parameters. This makes the method less modular than suggested. Specifically, Table 4 does not fully reveal the isloated impact of eac hmodule, and the hyper-parameters settings in Table 5 lack sufficient justification for their chosen values. A more thorough study on component interactions and hyper-parameter sensitivity would strengthen the work, particularly since reward-guided generation is known to be sensitive to reward scale and optimization dynamics [1].
>
> Thanks for your comments.
>
> ### 1. **Clarification on the Eight Hyperparameters**
>
> > the actual pipeline consists of multiple components and eight hyper-parameters
>
> We believe the eight hyperparameters you are referring to come from **Tab. 5**. However, only two of these hyperparameters—**n** and $\delta$ —belong specifically to our method. The other hyperparameters, such as **eta**, **w**, and **mp**, are general diffusion hyperparameters common to all DDIM-based methods. The remaining such as **m_dymo**, **m_dno**, and **n_das**, are from the baselines we compared against. Therefore, it cannot be said that our method is not modular.
>
> ### 2. **Isolated Impact of Each Module**
>
> > Table 4 does not fully reveal the isolated impact of each module
>
> In fact, we have conducted **ablation experiments** based on the **Diversity-Fidelity** and **Efficiency-Quality** trade-offs that we proposed. These experiments, presented in a progressive manner, demonstrate the effectiveness of each component. Additionally, in **Fig. 6**, we provide detailed experiments on the intensity of these two trade-offs, which we believe fully analyze the dynamic nature of our method.
>
> ### 3. **Justification for Hyperparameter Settings**
>
> > hyper-parameters settings in Table 5 lack sufficient justification for their chosen values
>
> For the reasons of selecton:
> - **n** is chosen based on aligning the **nFE** across different methods, ensuring that the evaluation is consistent.
> - **epsilon** serves as the threshold to determine whether the value approaches zero.
>
> For all baselines, we have followed their **official default settings** wherever possible, ensuring fair comparison.
>
> Furthermore, we have included **Appendix C.4**, where we provide further explanation regarding the performance of different hyperparameter selection strategies (e.g., **n = 2, 4, 8, 16**). We hope this additional explanation addresses the concerns and supports the chosen parameters in line with your high standards.

---

> ### Author Response · Authors · 2025-11-24
> **Point-by-Point Response to the Esteemed Reviewer QPUb (part 2)**
>
> # Weakness 3
> > The paper does not compare against recent fine-tuning based alignment approaches such as [2] DRaFT, [3] Diffusion-RPO, and [4] DRTune. Adding these baselines would better contextualize the claimed advantages in quality and efficiency compared to fine-tuning based method.
>
> Thank you for your insightful comments. We are grateful for your thoughtful suggestions and have made appropriate revisions in the updated PDF. Below, we provide a point-by-point response to address your concerns and further clarify the differences between our work and the related works you mentioned.
>
> ### 1. **Discussion with Related Works**
>
> First, we would like to clarify that the works you mentioned, **[2] DRaFT**, **[3] Diffusion-RPO**, and **[4] DRTune**, are all **training-based methods**, not **training-free methods**. Hence, the biggest difference between our work and these methods is the proposed LitExplorer is a **training-free method*, which conducts guidance during the inference time. We agree that discussing these methods can help improve and improve our work. Hence, we appreciate your suggestion and have incorporated relevant insights from these works to enhance our understanding and approach.  have added proper citations and comparative discussions in the **Related Work** section of the paper, addressing the works you mentioned.
>
> ### 2. **Experimental Comparison**
> We have **carefully reproduced** the **DRaFT** approach and added the comparison results to **Tab. 1** in the updated PDF.
> Regarding **Diffusion-RPO** and **DRTune**, although we were unable to find the official code for reproduction, we have included the latest inference-time baselines: **tt-scale** [1] and **FK-steer** [2], both of which provide state-of-the-art comparisons. The new results still demonstrate that **our method remains highly competitive**.
>
> We hope that these additions and clarifications will meet your high expectations.
>
> **References**:
> [1] Inference-time scaling for diffusion models beyond scaling denoising steps, CVPR, 2025
> [2] A General Framework for Inference-time Scaling and Steering of Diffusion Models, ICML, 2025
>
> ---
>
> # Weakness minor 1
> > The experiments do not present an execution time (or a wall-clock runtime) comparison on overhead breakdown for the full pipeline.
>
> Thank you for raising this issue. In **Appendix Fig. 10**, we provide a complete comparison of **wall-clock runtimes**. As shown, after integrating our method as a plugin to the base method, we observe a significant reduction in **wall-clock runtime**. We hope this addresses your concerns.
>
> ---
>
> # Weakness minor 2
> > 2.The experiments primarily focus on reward-model metrics without human preference studies or user evaluation. While these scores improve, it remains unclear whether the gains reflect real perceptual quality rather than reward-model alignment. Incorportaing even a small-scale human evaluation would make the empirical validation better.
>
> Thank you for your question. We have conducted human visual experiments based on comparisons. Specifically, we recruited 5 participants with no prior knowledge to make preference selections between image pairs before and after the addition of **Ours**. The prompts were provided alongside the image pairs as references.
>
> As shown in **Appendix C.6**, our method demonstrated a high **human evaluation win rate**, which validates the superiority of our method when used as a plugin.
>
> ---
>
> # Weakness minor 3
> > The paper shows experimental results on U-Net based models(SDv15 and SD XL1.0), and it would be better to explore its applicability to recent Transformer-based models such as FLUX or SD3.
>
> We have provided the experiments on SD3 as you requested. The results are shown in **Fig. 13** and **Tab. 11**, and the analysis is detailed in **Appendix C.8**.

---

> ### Author Response · Authors · 2025-11-24
> **Point-by-Point Response to the Esteemed Reviewer QPUb (part 3)**
>
> # Question 1
> >  Diffusion sampling assumes that intermediate states lie approximately on a perturbed noise manifold with respect to the forward process [5], [6]. Since the method introduces additional gradients and regularization signals during sampling, could these guidance terms push intermediate latents off the diffusion trajectory? How does the regularizations work within this method in terms of preverving manifold during denoising process.
>
> We would like to address your concerns from the following angles:
>
> ### 1. **On the Manifold Shift and Its Justification**
>
> Your understanding of the Manifold shift is quite valid. In fact, most fine-tuning methods do not maintain a "classic" diffusion manifold after the method is applied. However, our primary focus is on **generating better results**, rather than strictly preserving the original diffusion trajectory. As demonstrated by existing RL-based methods, such considerations are often not taken into account. For instance, methods like **ReFL (Image Reward)** [1], **DRaFT** [2], **Diffusion-DPO** [3], and **SPO** [4] focus on improving generation quality without worrying about Manifold preservation.
>
> ### 2. **Preserving the Classic Manifold in Some Methods**
>
> Some algorithms do attempt to preserve the classic diffusion Manifold, such as **Adjoint Matching** [5]. However, this comes at a high computational cost, as it requires the **JVP framework** for gradient backpropagation on the score function during rollout states.
>
> ### 3. **Regularization to Balance the Shift**
>
> We introduced **regularization** as a balancing term to explicitly encourage new manifold to avoid deviating too far from the original one. This helps prevent the collapse of the entire generation process. However, we do not strictly require **manifold preservation**—our goal is to maintain high-quality generation, and the regularization helps achieve this without forcing the model to strictly preserve the classic diffusion trajectory.
>
> ### 4. **Empirical Evidence of Final Generated Manifold Similarity**
>
> Experimentally, we demonstrate in **Fig. 2** and **Fig. 12** that the generated distribution of our method is closer to that of classic diffusion, indicating that our generated manifold are indeed closer to the original ones.
>
> [1] Imagereward: Learning and evaluating human preferences for text-to-image generation
>
> [2] Directly fine-tuning diffusion models on differentiable rewards
>
> [3] Diffusion model alignment using direct preference optimization
>
> [4] Aesthetic Post-Training Diffusion Models from Generic Preferences with Step-by-step Preference Optimization
>
> [5] Adjoint Matching: Fine-tuning Flow and Diffusion Generative Models with Memoryless Stochastic Optimal Control
>
> ---
>
> # Question 2
> > [Typos]  in Line 226 and in Equation (7) are not clealy distinguished. Also, in line 299 and  in Figure 3 should be unified.
>
> Thank you for your comments.
>
> ### 1. **Regarding Line 226 and Eq. 7:**
>
> Based on the definition in **Eq. 6**, we can see that $\mathbf{x}\_t\^\*$
> represents the original latent at step $t$, while $\mathbf{x}_{t|0}^*$ represents the latent at step $t$directly predicted from $x_0$. Specifically, we have:
>
> $\mathbf{x}\_{0|t} = f\_\theta\(\mathbf{x}\_t, t\)$
>
> This process is explained in **line 164** of the main text.
>
> We hope this clarifies the distinction between **Line 226** and **Eq. (7)**. If you have any further questions, we are happy to discuss them.
>
> ### 2. **Regarding Line 299 and Fig. 3:**
>
> We apologize for the oversight regarding **Line 299** and **Fig. 3**. We have now updated the PDF to unify and address this issue.

---

> ### Author Response · Authors · 2025-11-25
> **Update of results on the advanced SD3.**
>
> Dear reviewer,
>
> Based on your valuable suggestion and guidance, we have **completed the reproduction and comparison** of our method and the baselines on the **SD3** model. We firmly believe that this experiment is crucial for enhancing the quality of our paper.
>
> Please refer to the updated PDF in **Appendix Sec C.8**. In this section, we first quantitatively evaluate the reward learning process (**Tab. 11**) and provide a multi-scale image quality assessment. Our method consistently maintains the conclusions drawn from previous experiments, demonstrating that **LitE can effectively fit the reward while simultaneously optimizing and maintaining image diversity and overall quality**.
>
> Additionally, the visual results in **Fig. 13** further confirm the superiority of our aesthetic optimization.

---

> > ### Comment · Reviewer_QPUb · 2025-11-25
> > **Response to rebuttal**
> >
> > Thank you for your detailed answers.
> > I think the authors have strengthened the paper by adding several new experiments: DRaFT comparisons and SD3 results. These collectively reinforce the empirical effectiveness of the method.
> >
> > However, two concerns remain. First, although the authors describe the Control Network as a pretrained module consistent with a training-free framework, it is still unclear how this network was actually obtained. The paper does not specify its training details, dataset composition, scale, or generalization behavior, making it difficult to assess whether it can be regarded as a broadly available pretrained component in the same sense as models like PickScore. Table 10 hints that the network might be trained on a “simple animal” dataset; if this is indeed the dataset used in the final method, clarification is needed regarding why this dataset was chosen and how its distribution influences generalization to arbitrary prompts. Without this information, the validity of the training-free claim remains ambiguous. Second, the explanation of manifold preservation is still incomplete. Since the method introduces additional guidance and regularization terms during sampling, it is not yet clear how these signals interact with the diffusion trajectory or whether the resulting latents remain aligned with the underlying manifold. I would appreciate further clarification on how the authors interpret such deviations and how they understand the role of their regularization in this context.

---

> > > ### Author Response · Authors · 2025-11-25
> > > **Response to your further questions. (Part 1 for Q1)**
> > >
> > > We sincerely thank you for your timely reply and recognition of our rebuttal. We are also very pleased that you raised two further valuable questions. Here, we will provide detailed answers to each of them.
> > >
> > > ---
> > >
> > > # Further Question 1:
> > > > Athough the authors describe the Control Network as a pretrained module consistent with a training-free framework, it is still unclear how this network was actually obtained. The paper does not specify its training details, dataset composition, scale, or generalization behavior, making it difficult to assess whether it can be regarded as a broadly available pretrained component in the same sense as models like PickScore. Table 10 hints that the network might be trained on a “simple animal” dataset; if this is indeed the dataset used in the final method, clarification is needed regarding why this dataset was chosen and how its distribution influences generalization to arbitrary prompts. Without this information, the validity of the training-free claim remains ambiguous.
> > >
> > > ## 1. **Detailed Training Process and Explanation**
> > >
> > > We start by reporting and explaining our training process as follows:
> > >
> > > 1. **Given a simple prompt set \( S \), e.g., the simple animal prompt set that we used**
> > >    **Explanation**: The simple animal set is commonly used in RL fine-tuning methods like **DDPO**, **DPOK**, and **TDPO**. We chose this set for the following reasons:
> > >    - It has been widely used in well-known methods and is recognized by the community.
> > >    - It is simple, which makes training easier, and it further helps demonstrate that the noise-prediction model can generalize well, as we can achieve generalization to more complex datasets with such a small training set.
> > >
> > > 2. **Obtain prompt $p$ from $S$), and have the SD model generate a series of denoised intermediate latents for this prompt: $Z = \{z_{50}, ..., z_0\} = G(p)$, where we assume 50 denoising steps.**
> > >    **Explanation**: Using the SD model, we can obtain a complete denoising trajectory for p, where $Z=\{z_{50},  ..., z_0\}$ represents the trajectory from pure noise to a clear state.
> > >
> > > 3. **Let the Control Network predict for Z, resulting in a set of prediction confidence values $C = \{c_{50}, ..., c_0\}$ between 0 and 1.**
> > >    **Explanation**: The Control Network is a simple 4-layer convolutional network. The input shape is the same as z, and the output, after passing through a sigmoid function, is constrained to the range of 0 to 1. This can be viewed as a binary classification task.
> > >
> > > 4. **Assign ground-truth labels to the set Z , with the first n elements assigned label 0 (i.e., noisy), the last n assigned label 1 (i.e., clean), and the middle \( 50 - 2n \) elements left unlabeled (i.e., unsupervised). The labels are used to compute cross-entropy loss with \( C \) and optimize the Control Network's weights.**
> > >    **Explanation**: In practice, we set \( n = 7 \), and expect that the middle \( 50 - 2n \) steps, which are unsupervised, will implicitly learn the continuous transition from 0 to 1.
> > >
> > > This describes the detailed **Control Network's training process**. Based on this, we can further explain **Table 10**:
> > >
> > > ## 2. **Interpretation of Table 10**
> > >
> > > From **Tab. 10**, we observe the following after training on the very simple **simple-animal** dataset:
> > >
> > > 1. The model learned the expected **implicit continuous effect**, recognizing that the last 12.7 steps of the denoising trajectory are considered clean (i.e., denoise complete), which aligns with how denoising is typically perceived (e.g., as in DyMo). In the later stages of denoising, changes become more subtle.
> > >
> > > 2. The **generalization results** on other datasets demonstrate the model's versatility. Despite training on a simple dataset, the model maintains similar levels of denoising perception on more complex datasets like **Pick-a-pic**, **HPSv2**, and **GenEval**, showing that the model can be treated as a **pretrained plugin**.
> > >
> > > ## 3. **Why the Control Network can be Simple Yet Effective**
> > >
> > > The effectiveness of the **Control Network** lies in the nature of DDIM-based denoising settings. Specifically, for the SD model, the noise level of each intermediate state is already roughly determined by a set of hyperparameters in the **DDIM-scheduler**, such as fixed standard deviations for each step. These parameters are set based on the denoising steps, and thus the noise levels for intermediate states are coarsely pre-defined, independent of the prompt.
> > >
> > > Given this predefined range, our network is able to learn and generalize more easily. The main function of the network is to predict the noise **fine-grained and dynamically**, refining the predictions based on the coarse granularity provided by the scheduler. This allows the Control Network to perform effectively in terms of both efficiency and generalization.
> > >
> > > ---
> > >
> > > We hope this detailed explanation clarifies the process and the effectiveness of our approach. Please feel free to reach out if there are any further questions.

---

> ### Author Response · Authors · 2025-11-25
> **Response to your further questions. (Part 2 for Q2)**
>
> # Further Question 2
> > Second, the explanation of manifold preservation is still incomplete. Since the method introduces additional guidance and regularization terms during sampling, it is not yet clear how these signals interact with the diffusion trajectory or whether the resulting latents remain aligned with the underlying manifold. I would appreciate further clarification on how the authors interpret such deviations and how they understand the role of their regularization in this context.
>
> Thank you for your insightful question.
>
> ## 1. **Our Interpretation to Manifold Issue**
>
> We have indeed been paying close attention to the manifold hypothesis. However, we believe that the manifold impact is more significant in the **image space** than in the **latent space** where our algorithm operates. Specifically, in the latent space, only **more aggressive manifold shifts** are likely to cause catastrophic results.
>
> A strong supporting argument for this is found in **JIT [1]**, where direct image prediction methods are more efficient when considering the manifold. However, this effect is less pronounced in latent diffusion models.
>
> ## 2. **Additional Measures to Prevent Aggressive Manifold Shifts**
>
> Moreover, our framework includes two key features to further prevent aggressive manifold shifts:
>
> 1. **L2 Loss for Encouraging Consistency**: We use an **L2 loss** to encourage the relative consistency of the new trajectory with the original trajectory, thereby reducing the risk of deviation. This aligns with the **Diversity-Fidelity tradeoff** we proposed in the paper.
>
> 2. **Filtering Out of Extreme Deviations**: If the model does indeed deviate from the data manifold, the reward for such data is naturally not high. We effectively filter out these outliers. Our experimental results in **Tab. 1** show that the images generated by our method are naturally diverse and receive high rewards. This strongly supports the idea that our method remains within the **proper data manifold**.
>
> We hope this provides a clearer understanding of our approach and addresses the concerns regarding manifold issue.
>
> [1] Li T, He K. Back to Basics: Let Denoising Generative Models Denoise

---

### Official Review · Reviewer_6kFr · 2025-11-04

**Soundness:** 3
**Presentation:** 2
**Contribution:** 2
**Rating:** 4
**Confidence:** 3

**Summary:**

The paper proposes LitExplorer, a training-free plugin to align diffusion models with target rewards while preserving diversity and reducing compute. It introduces an Inheritance-Restart exploration mechanism with an adaptive diversity-fidelity controller, and a Quality-Efficiency arbitration that screens ineffective guidance and triggers dynamic early stopping based on marginal reward gains. Experiments on SD v1.5 and SD-XL across Pick-a-Pic and HPSv2 show improved alignment preference, fidelity, and diversity, with faster sampling.

**Strengths:**

- Clear motivation; mitigates distribution narrowing and reward hacking while maintaining fidelity.
- Performance gains: top or top-2 on most of 12–13 metrics across datasets and backbones, improved diversity (LPIPS/TCE/NIQE), and generalizes to multiple reward objectives (PickScore, AES, ImageReward) as a drop-in plugin.
- Efficiency gains; reduces compute without degrading quality.

**Weaknesses:**

Empirical Support for Motivation
- DAS claims to avoid reward over-optimization while allowing reward guidance at inference time. However, it is unclear whether Fig. 1 includes results for other baselines like DAS beyond DyMO. It would strengthen the argument to include such comparisons.

Novelty
- The proposed _guidance screening_ module appears similar to early stopping. It seems to be an independent component that could be easily integrated into other methods rather than a fundamentally novel contribution. While tackling efficiency is valuable, the novelty of introducing a standalone early-stopping-like mechanism is questionable.


Experiment Setup
- The experiments only cover SD1.5 and SDXL; evaluations on more recent backbones would improve credibility.
- Recent diffusion inference-time scaling baselines [1, 2] are missing.
- Since the performance of sampling-based methods (DAS, [1], [2]) depends heavily on hyperparameters such as the number of particles, while their configurations used were missing. Experiments analyzing scaling with respect to nFE or similar metrics are also essential.

---

[1] Inference-time scaling for diffusion models beyond scaling denoising steps, CVPR, 2025

[2] A General Framework for Inference-time Scaling and Steering of Diffusion Models, ICML, 2025

**Questions:**

See weaknesses above

---

> ### Author Response · Authors · 2025-11-24
> **Point-by-Point Response to the Esteemed Reviewer 6kFr (part 1)**
>
> Dear reviewer  6kFr,
> We are deeply grateful for the time you have dedicated to reviewing our work and for your insightful suggestions. We are also encouraged by your recognition of our motivation and the gains we have achieved. Based on your feedback, we have carefully optimized the PDF, with the modifications highlighted in *italicized blue* for clarity. Below, we provide a point-by-point response to all of your queries.
>
> ---
>
> # Weakness 1
> > DAS claims to avoid reward over-optimization while allowing reward guidance at inference time. However, it is unclear whether Fig. 1 includes results for other baselines like DAS beyond DyMO. It would strengthen the argument to include such comparisons.
>
> Thank you for your comment. Firstly, we noticed that **Fig. 1** does not include any comparative experimental design but instead focuses on subjective results. Therefore, we assume that you might be referring to the **distribution visualization experiment** in **Fig. 2**. If our understanding is incorrect, we sincerely apologize and would appreciate it if you could kindly point it out.
>
> Based on this understanding, we sincerely thank you for highlighting this important experiment. We completely agree that **DAS**, a sampling-based method, may exhibit differences in distribution performance compared to **DyMO**. To make a clearer comparison between **Ours** and **DAS**, and to avoid excessive data points that could hinder the readability of the visualization, we have added a **DAS comparison experiment** in **Appendix C.7** of the newly uploaded PDF.
>
> As shown in the updated results, **DAS** indeed shows more sparse distribution compared to **DyMO** (as quantified by the silhouette score). However, our method still demonstrates superior performance.
>
> We hope this clarifies the concern and we truly appreciate your valuable input.
>
> ---
>
> # Weakness 2
> > The proposed guidance screening module appears similar to early stopping. It seems to be an independent component that could be easily integrated into other methods rather than a fundamentally novel contribution. While tackling efficiency is valuable, the novelty of introducing a standalone early-stopping-like mechanism is questionable.
>
> ## **First, the novelty of guidance screening**
>
> - **A as Guidance Screening**
> - **B as Common Early Stopping Mechanism**
>
> #### 1. **Difference in Stopping Target (A. Guidance vs B. Stopping Entire Diffusion)**
>
> - **Traditional early stopping** halts the **entire sampling process** when a certain condition is met.
> - **Our method** never stops the sampling process. Instead, it only stops ineffective reward guidance, while the diffusion model continues to execute the full denoising trajectory.
> Therefore, our approach is fundamentally **guidance scheduling**, not early stopping. It is a **step-wise guidance scheduling mechanism** rather than a crude **trajectory truncation** method.
>
> #### 2. **Difference in Handling Logic (Local Filtering vs Global Early Termination)**
>
> - **Traditional early stopping** is a form of **“early termination”**:
>   - Logic: Once the condition is met → Stop all steps → No further reasoning is performed.
> - **Our method** uses **signal filtering**:
>   - Logic: Evaluate each step → If no gain, discard the step’s guidance → Continue sampling.
>
> #### 3. **Difference in Decision Trigger Condition Space (Dual-condition vs Single Threshold)**
>
> - **Traditional early stopping** typically uses a **single threshold** rule (e.g., no loss decrease, fixed number of steps).
> - **Our method** triggers based on a **2D state space** condition:
> $p\_t = 1 \quad \text{and} \quad \Delta r_t \le \delta\_r$
> Both conditions must be met:
> - Sufficient noise progress
> - Saturation of reward marginal gain
> Only then will the guidance be stopped.
>
> Therefore, our method uses a **strongly coupled binary state decision**, while early stopping relies on a single threshold.
>
> ## **Second: the Connection to Our overall design**
>
> The **guidance screening** module is directly tied to the balance of **DIVERSITY** and **FIDELITY** in our method.
> Guidance screening dynamically adapts based on changes in $p_t$, which serves as a balance factor. When $p_t$ reaches 1, this means the dynamic balance factor concludes that further optimization is unnecessary.
>
> Therefore, **guidance screening** is essentially an **extreme case of our trade-off module**, and it is deeply connected to other aspects of our design.
>
> We hope this explanation clarifies the misunderstanding and provides a clearer picture of the unique contributions of our method.

---

> ### Author Response · Authors · 2025-11-24
> **Point-by-Point Response to the Esteemed Reviewer 6kFr (part 2)**
>
> # Weakness 4
> > Recent diffusion inference-time scaling baselines [1, 2] are missing.
>
> Thank you for your comment. We have reproduced the results of **advanced TT-scale** [1] (https://github.com/sayakpaul/tt-scale-flux) and **FK-steer** [2] (https://github.com/zacharyhorvitz/Fk-Diffusion-Steering) using the official code and default settings, and we report the results in **Tab. 1**. As shown, our method remains highly competitive compared to these state-of-the-art approaches.
>
> ---
>
> # Weakness 5
> > Since the performance of sampling-based methods (DAS, [1], [2]) depends heavily on hyperparameters such as the number of particles, while their configurations used were missing. Experiments analyzing scaling with respect to nFE or similar metrics are also essential.
>
> Thank you for your suggestion.
>
> First, we have already provided detailed baseline reproduction hyperparameters in **Appendix Tab. 5**. These hyperparameters were selected **based on** the number of iterations per step in DyMO and the particle count (2) used in **Ours**, as well as the average number of iterations in the inherited restart mechanism (2), aiming to keep them roughly consistent for fair comparisons.
>
> Regarding the **scaling experiments** you mentioned, we place great importance on this and have designed **nFE-based scaling experiments**. These experiments compare the fitting of the target (PickScore) and the changes in diversity (LPIPS). The experimental results show that iteration-based methods are more suitable for lightweight inference-time scaling, while sampling-based methods perform better when computational cost is not considered.
>
> Our method is designed to leverage both **sampling** and **optimizing** simultaneously:
> - using sampling to expand the exploration range
> - applying optimization to refine the reward target.
>
> Thus, we have designed two types of evaluations for LitExplorer's scaling nFE: increasing **sampling** or scaling **optimization iteration**. The final experimental results prove the **comprehensive superiority** of our approach.

---

> ### Author Response · Authors · 2025-11-24
> **Point-by-Point Response to the Esteemed Reviewer 6kFr (part 3)**
>
> # Weakness 3
> > The experiments only cover SD1.5 and SDXL; evaluations on more recent backbones would improve credibility.
>
> We greatly appreciate your suggestion. We are **currently working on reproducing our method on the more advanced and completely different DiT-based SD3 architecture**. We will complete the experiments as soon as possible and report the results to you. Thank you so much for your patience.

---

> ### Author Response · Authors · 2025-11-25
> **Update of results on the advanced SD3.**
>
> Dear reviewer,
>
> We are pleased to report that we have **completed the reproduction and comparison** of our method and the baselines on the **SD3** model. We firmly believe that this experiment is crucial for enhancing the quality of our paper.
>
> Please refer to the updated PDF in **Appendix Sec C.8**. In this section, we first quantitatively evaluate the reward learning process (**Tab. 11**) and provide a multi-scale image quality assessment. Our method consistently maintains the conclusions drawn from previous experiments, demonstrating that **LitE can effectively fit the reward while simultaneously optimizing and maintaining image diversity and overall quality**.
>
> Additionally, the visual results in **Fig. 13** further confirm the superiority of our aesthetic optimization.

---

### Author Response · Authors · 2025-12-03
**Overall Comment (3): Point-by-Point Response List for AC**

Below is a refined version that explicitly reflects this:

---

# Summary of Actions and Responses to Major Concerns

We summarize below how each concern has been addressed, including **new experiments, clarifications, and revisions**.

---

## Reviewer: 6kFr

### W1: “DAS also claims to avoid reward over-optimization… include distribution comparisons with DAS like Fig. 2.”
**Action:**
- Added **distribution comparisons with DAS**
- New experiment: **Fig. 12**
- Added analysis: **Appendix C.7**
**Response:** part1_w1


### W2: “The proposed guidance screening module appears similar to early stopping.”
**Action:**
- Added conceptual clarification in the text
- Explicitly distinguished guidance screening from early stopping
**Response:** part1_w2


### W3: “Evaluations on more recent backbones.”
**Action:**
- Added experiments on **SD3**
- New section: **Appendix C.8**
**Response:** part1_w3


### W4: “Recent two baselines [1,2].”
**Action:**
- Implemented and added **TT-scale [1]** and **FK-steer [2]**
- Updated results in **Tab. 1**
**Response:** part2_w4


### W5: “Baseline hyperparameters.”
**Action:**
- Clarified Tab. 5
- Added **nFE scaling experiment (Fig. 9)**
**Response:** part2_w5

---

## Reviewer: QPUb

### W1: “Claims training-free but uses a learned Control Network.”
**Action:**
- Added explanation in **Appendix C.3**
- Added generalization experiment: **Tab. 10**
**Response:** part1_w1


### W2: “Pipeline has multiple components and eight hyperparameters…”
**Action:**
- Updated **Tab. 5** (highlighting parameters)
- Added sensitivity study: **Fig. 9**
- Added explanation: **Appendix C.4**

**Clarification:**
Only **n** and **δ** are specific to our method. Other parameters are diffusion defaults or baseline settings.

**Response:** part1_w2


### W3: “No comparison to DRaFT, Diffusion-RPO, and DRTune.”
**Action:**
- Added available comparison in **Tab. 1**
- Included literature discussion in **Appendix A**
**Response:** part2_w3


### Minor Weakness 1: Wall-clock runtime missing
**Action:**
- Added full wall-clock analysis: **Fig. 10**


### Minor Weakness 2: Human preference evaluation missing
**Action:**
- Added human study: **Fig. 11**
- Added discussion: **Appendix C.6**

### Minor Weakness 3: Only U-Net backbones
**Action:**
- Added **SD3 experiments**: Fig. 13, Tab. 11
- Added analysis: **Appendix C.8**

---

## Reviewer: 6nmv

### W1: “Design choices not clearly justified.”
**Action:**
- Expanded discussion and added experiments
**Response:** part1_w1


### W2: “No margin of error.”
**Action:**
- Added CoV analysis: **Tab. 9**
**Response:** part1_w2


### W3: “Not training-free due to Control Network.”
**Action:**
- Added explanation + generalization experiment
- Appendix C.3, Tab. 10
**Response:** part1_w3


### Q1: “Reward selection reduces diversity.”
**Action:**
- Added distribution analysis: **Fig. 2, Fig. 12**
- Extended metrics in **Tab. 1–3**
**Response:** part1_Q1

### Q2: “Inheritance motivation unclear.”
**Action:**
- Added time/quality ablation: **Tab. 4**
**Response:** part2

---

# Conclusion

In response to reviews, we took the following **concrete actions**:

- Added **new baselines**
- Added **new backbones (SD3)**
- Added **distribution analysis**
- Added **human evaluation**
- Added **wall-clock timing**
- Added **hyperparameter study**
- Added **variance analysis**
- Clarified **training-free design**
- Expanded **theoretical justification**
- Updated **related work**

We believe the revision is now substantially strengthened both empirically and conceptually.

---

---

### Author Response · Authors · 2025-12-03
**Overall Comment (2): Experiment List for AC**

# Experiment List

⚠️  Dear AC, Since we revised the paper multiple times in response to reviewer comments and discussions, the numbering of figures and tables has changed compared to the original comments and rebuttal messages. We kindly ask you to refer to **this list** as the authoritative mapping between experiments (Fig. Sec. Tab.) and reviewer queries. Thank you for your understanding.


1. **Generated Images of LitExplorer** — *Fig. 1*
   Visual impression demonstrating the overall superior quality of our method.

2. **Comparison of Visualized Latent-Space Distributions** — *Fig. 2, Fig. 12*
   Analysis of maintaining distribution diversity after guidance.
   *(6kFr Weakness 5)*

3. **Comprehensive Metrics on SD-v1.5 (PickScore, DRaFT, TT-Scale, FK-Steer)** — *Tab. 1*
   Demonstrates superior overall generative quality.
   *(6kFr Weakness 4, QPUb W3)*

4. **Evaluation on Additional Dataset (HPSv2)** — *Tab. 2*
   Demonstrates method generality.

5. **Evaluation on More U-Net Backbones (SD1.4, SD2.1, SDXL)** — *Tab. 3, Tab. 6, Tab. 7*
   Demonstrates method generality.

6. **Evaluation on DiT-Based Backbone (SD3)** — *Tab. 11, Fig. 13*
   Further verifies applicability and scalability.
   *(6kFr W3, QPUb Minor Weakness 3)*

7. **Plug-and-Play Effectiveness Across Different Base Methods** — *Fig. 4*
   Verifies plugin capability from five perspectives.

8. **Ablation Study** — *Tab. 4*
   Demonstrates the interplay and effectiveness of each proposed component.

9. **Qualitative Results on Image Diversity** — *Fig. 5*
   Shows diversity in color, posture, and semantics.

10. **Intensity Parameter Analysis** — *Fig. 6*
    Compares adaptive diversity–fidelity trade-off with fixed exploration and regularization strengths.

11. **Mechanism Analysis** — *Tab. 8*
    Shows denoising steps and restart counts under different settings.

12. **Main Visual Comparisons** — *Fig. 7*
    Qualitative comparison on photorealistic and surreal prompt sets.

13. **Results on Additional Optimization Target (Aesthetic Score)** — *Fig. 8*
    Demonstrates method generality.

14. **Coefficient of Variation (CoV) Analysis** — *Tab. 9*
    Validates statistical significance of the results.
    *(6nmv W2)*

15. **Generalization of Step Control Network** — *Tab. 10*
    Verifies cross-dataset generalization and supports the training-free claim.
    *(QPUb Weakness 1, 6nmv W3)*

16. **Performance–Computation Trade-off** — *Fig. 9*
    Generation quality under different nFE levels.
    *(QPUb W2)*

17. **Detailed Wall-Clock Runtime Breakdown** — *Fig. 10*
    Component-wise time cost and practical deployment potential.
    *(QPUb Minor Weakness 1)*

18. **User Study (Human Evaluation)** — *Fig. 11*
    Subjective quality assessment by human raters.
    *(QPUb Minor Weakness 2)*

19. **Diversity Within a Single Denoising Trajectory** — *Fig. 14*
    Analysis of exploration induced by inference-time scaling.

---

### Author Response · Authors · 2025-12-03
**Overall Comment (1): Summary for AC**

# Summary

**All comments have been sufficiently addressed.**

Dear Area Chair,

We respectfully provide a concise summary of our rebuttal updates to help save your time. In short, we are confident that **all reviewers' concerns have been fully addressed without omission** (see the attached issue-resolution index and answer mapping prepared specifically for you).

---

## 1. Experimental Updates

Building upon the original **16 experiments** in the paper, we **added 5 new independent experiments**, bringing the total to **19 experiments**. Key additions include:

- **(a) Evaluation on advanced DiT-based SD3**: *Tab. 11, Fig. 13*
- **(b) Expanded baselines**, including **DRaFT, TT-Scale, FK-Steer**: *Tab. 1*
- **(c) Statistical significance analysis (CoV)**: *Tab. 9*
- **(d) Comprehensive computational cost analysis**: *Fig. 9, Fig. 10*

For the complete list of all experiments and the corresponding addressed comments, please refer to **Overall Comment 2 — Experiment List**. These additions address all experimental requests raised by the reviewers.

---

## 2. Conceptual and Theoretical Clarifications

Beyond the paper’s original multi-objective design and core contributions, we provided further clarifications on novelty, theory, and design details. Highlights include:

- **(a) How exploration supplement increases trajectory diversity**
  See: *Point-by-Point Response to Reviewer 6nmv — part1_Q1*

- **(b) The role of inheritance-restart**: reuse reduces compute when optimization directions are similar
  See: *Reviewer 6nmv — part2_Q2*

No additional design objections remain unaddressed. A complete directory of all point-by-point responses is provided in **Overall Comment 3 — Point-by-Point Response List**, enabling precise lookup of each question and response.

---

## 3. Reviewers’ Positions After Rebuttal

We summarize each reviewer’s stance following our revisions:

- **6kFr**
  Recognized our motivation and experimental results. We added all requested experiments, including SD3, new baselines, and DAS distribution analysis with theoretical discussion. All points were addressed.

- **QPUb**
  Acknowledged our motivation and expressed *initial satisfaction* with the additional experiments (SD3 and advanced baselines). We further resolved concerns regarding the Control Network and manifold shift with detailed analysis.

- **6nmv**
  Recognized our comprehensive evaluation across multiple metrics. All raised concerns (Control Network, statistical significance, etc.) received detailed responses. While no follow-up was received, we believe all issues were thoroughly addressed.

---

## Conclusion

We are confident that:

- All reviewer concerns have been addressed comprehensively,
- QPUb has already shown initial satisfaction with our added experiments,
- Our expanded experimental and theoretical clarifications resolve remaining ambiguities.

Should you have any further questions, we would be pleased to discuss them in detail. We next provide the **Detailed Experiment List** and the **Point-by-Point Response Index** for your convenience.

Sincerely,
The Authors

---

### Meta-Review · Area_Chair_Bat5 · 2025-12-26

**Summary:**

This paper received mixed reviews. While several reviewers identified some merits, most raised substantive concerns. Although the authors attempted to address these issues, many of the key concerns remain unresolved.

**Reviewer Concerns:**

While the authors have made an effort to address reviewer questions, several fundamental concerns remain unresolved, and the rebuttal was not convincing to the reviewers. The proposed approach largely relies on heuristic modifications to existing training-free diffusion guidance, rather than introducing principled modeling or learning advances. As such, the contribution feels incremental and engineering-driven.

More importantly, the paper does not adequately address memorization and bias, which are well-known and critical issues in diffusion models. It remains unclear whether the proposed method genuinely improves generalization or whether it biases generation toward a narrow subset of the pretrained distribution. There is no analysis of whether the generated samples simply replicate training data characteristics, amplify existing dataset biases, or collapse toward high-reward but low-diversity modes.

Overall, despite promising empirical results, the lack of theoretical grounding, insufficient analysis of bias and memorization, and reliance on heuristics prevent the work from meeting the bar for acceptance at ICLR.

**Reviewer Scores:**

Based on the discussion so far and the number of significant concerns raised by the reviewers, it appears very unlikely that the reviewers would change their scores.

---

### Decision · Program_Chairs · 2026-01-26

Reject